# Implicit Bias of Adverarial Training for Deep Neural Networks

**Bochen Lyu**
DataCanvas Lab
DataCanvas, Beijing, China
lvbc@zetyun.com

**Zhanxing Zhu**
The University of Edinburgh, UK
zhanxing.zhu@gmail.com

## Abstract

We provide theoretical understandings of the implicit bias imposed by adversarial training for homogeneous deep neural networks without any explicit regularization. In particular, for deep linear networks adversarially trained by gradient descent on a linearly separable dataset, we prove that the direction of the product of weight matrices converges to the direction of the max-margin solution of the original dataset. Furthermore, we generalize this result to the case of adversarial training for non-linear homogeneous deep neural networks without the linear separability of the dataset. We show that, when the neural network is adversarially trained with $\ell_2$ or $\ell_\infty$ FGSM, FGM and PGD perturbations, the direction of the limit point of normalized parameters of the network along the trajectory of the gradient flow converges to a KKT point of a constrained optimization problem that aims to maximize the margin for adversarial examples. Our results theoretically justify the longstanding conjecture that adversarial training modifies the decision boundary by utilizing adversarial examples to improve robustness, and potentially provides insights for designing new robust training strategies.

## 1 Introduction

Deep neural networks (DNNs) have achieved great success in many fields such as computer vision (Krizhevsky et al., 2012; He et al., 2015) and natural language processing (Collobert & Weston, 2008) and other applications. These breakthroughs also lead to the importance of research on the robustness and security issues of DNNs, among which those about adversarial examples are especially prevalent. Adversarial examples are obtained by adding craftily designed imperceptible perturbations to the original examples, which can sharply change the predictions of the DNNs with high confidence (Szegedy et al., 2014; Nguyen et al., 2015). Such vulnerability of DNNs renders security concerns about deploying them in security-critical systems including vision for autonomous cars and face recognition. It is therefore crucial to develop defense mechanisms against these adversarial examples for deep learning.

To this end, many defense strategies have been proposed to make DNNs resistant to adversarial examples, such as adding a randomization layer before the input to the classifier (Xie et al., 2018), input transformations (Guo et al., 2018), adversarial training (Madry et al., 2018), etc. However, Athalye et al. (2018) pointed out that most of the defense techniques are ineffective and give a false sense of security due to obfuscated gradients except for adversarial training—a method which has not been comprehensively attacked yet. Given a $C$-class training dataset $\{(x_i, y_i)\}_{i=1}^n$ with natural example $x_i \in \mathbb{R}^d$ and corresponding label $y_i \in \{1, ..., C\}$, adversarial training is formulated as solving the following minimax optimization problem

$$\min_W \tilde{\mathcal{L}}(W) = \min_W \frac{1}{n} \sum_{i=1}^n \max_{\delta_i \in \mathcal{B}(0,\epsilon)} \ell\left(x_i + \delta_i(W), y_i; W\right), \tag{1}$$

where $f$ is the DNN function, $\ell$ is the loss function and $\delta_i(W)$ is the adversarial perturbation generated by some adversary $\mathcal{A}(x_i, y_i; W)$ typically depending on model parameters $W$, within the set $\mathcal{B}(0, \epsilon) = \{\delta : \|\delta\|_p \leq \epsilon\}$ around the natural example $x_i$. Commonly, adversarial training conducts a two-player game at each iteration: the inner maximization is to attack the model and

find the corresponding perturbation of the original example that maximizes the classification loss $\ell$; the outer minimization, on the other hand, is to update the model parameters $W$ with the gradient descent method such that the loss $\ell$ is minimized on adversarial examples generated by the inner maximization (see Algorithm 1 for details).

To have a better theoretical understanding of the fact that adversarial training empirically improves the robustness of DNNs against adversarial examples, Zhang et al. (2019) decomposed the prediction error for adversarial examples and identified a trade-off between robustness and accuracy while Li et al. (2020) studied the inductive bias of gradient descent based adversarial training for logistic regression on linearly separable data. Faghri et al. (2021) studied adversarial robustness of linear neural networks by exploring the optimization bias of different methods. Yu et al. (2021) studied adversarial training through the bias-variance decomposition and showed that its generalization error on clean examples mainly comes from the bias. However, even for the simplest DNN—the deep linear network, we notice that there exists no work to theoretically understand such robustness achieved by adversarial training through exploring its implicit bias.

On the other hand, for the standard training, recent works extensively explored the implicit bias imposed by gradient descent or its variants for DNNs in different settings. For a simple setting of linear logistic regression on linearly separable data, Soudry et al. (2018); Ji & Telgarsky (2018); Nacson et al. (2019b) derived that the direction of the model parameter converges to that of the max-$\ell_2$-margin with divergent norm. Shamir (2021) concluded that gradient methods never overfit when training linear predictors over separable dataset. Viewing the above model as a single-layer network, a natural but more complicated extension is that for the deep linear network on linearly separable data: Ji & Telgarsky (2019) proved that the gradient descent aligns the weight matrices across layers and the product of weight matrices also converges to the direction of the max-$\ell_2$-margin solution; Gunasekar et al. (2018) showed the implicit bias of gradient descent on linear convolution networks. Nacson et al. (2019a); Lyu & Li (2020) further promoted the study of implicit bias of gradient descent for standard training to the case of more general homogeneous non-linear neural networks and proved that the limit point of the optimization path is along the direction of a KKT point of the max-margin problem. Wei et al. (2019) studied the regularization path for homogeneous DNNs and also proved the convergence to the max-margin direction in this setting. Banburski et al. (2019) showed that gradient descent induces a dynamics of normalized weights converging to an equilibrium which corresponds to a minimal norm solution.

It is therefore our goal in this paper to theoretically understand the resistance to adversarial examples for adversarially trained DNNs, linear and non-linear ones, through the lens of implicit bias imposed by adversarial training. Due to the inner maximization, adversarial training differs a lot from standard training and one should be careful to analyze the perturbed training dynamics. For the adversarial training objective Eq. (1), various approaches have been proposed to solve the inner maximization, such as fast gradient sign method (FGSM (Goodfellow et al., 2015)) and its stronger version projected gradient descent (PGD (Madry et al., 2018)). The widely used adversarial training adopts PGD to attack the model, while recent work (Wong et al., 2020) also suggested that a weaker adversary can also surprisingly yield a model with satisfying robustness. Thus to conduct a comprehensive study about the implicit bias of adversarial training for DNNs, we will use $\ell_2$-fast gradient method (FGM (Miyato et al., 2016)), $\ell_\infty$ FGSM, $\ell_2$-PGD and $\ell_\infty$-PGD to solve the inner maximization of the adversarial training objective.

## 1.1 OUR CONTRIBUTION

In this paper, we devote to answering two questions. First, *is there any implicit bias imposed by adversarial training for DNNs without explicit regularization?* Second, if there exists such an implicit bias, *what are the convergence properties of the model parameters along the adversarial training trajectory?*

To this end, we first investigate the adversarial training with $\ell_2$ adversarial perturbations for deep linear networks on linear separable data where the allowed Euclidean distances from the adversarial examples to their corresponding original examples $\|x_i' - x_i\|_2$ are less than the max-$\ell_2$-margin of the original dataset. Despite the simplicity of this setting, this problem is meaningful due to its non-convexity and the introduction of adversarial examples, which heavily depend on the model parameters, during the training process. We prove that gradient descent for adversarial training

implicitly aligns weight matrices across layers and the direction of the product of weight matrices also surprisingly converges to that of the max-$\ell_2$-margin solution of the original dataset—similar to that of standard training for deep linear network (Ji & Telgarsky, 2019). Our results significantly generalize those in Li et al. (2020) for adversarial training of logistic regression. This simple yet insightful case positively answers our first question but partially answers the second one because it still remains unclear why such convergence property can improve robustness considering its similarity to that for the standard training. In fact, adversarial training differs from standard training in the way how they impose such convergence property for parameters: the first one is to maximize the margin of adversarial examples while the latter is maximizing that of the original dataset, and these two optimization problems happen to possess solutions along the same direction.

We then move forward to explore a more general situation, adversarial training for homogeneous non-linear DNNs *without* the linear separability of the dataset. We study the limit point of the normalized model parameters along the adversarial training trajectory and show that

**Theorem 1** (Informal). *When the deep neural network is adversarially trained with one of the $\ell_2$-FGM, FGSM, $\ell_2$-PGD and $\ell_\infty$-PGD perturbations, the limit point of the normalized model parameters is along the direction of a KKT point of a constrained optimization problem which aims to maximize the margin of adversarial examples.*

This indicates that adversarial training is implicitly maximizing the margin of adversarial examples rather than that of original dataset. Thus Theorem 1 provides another view for the high bias error on clean examples of adversarial training in Yu et al. (2021) since distributions of adversarial and clean examples are different. To the best of knowledge, these results are the first attempt to analyze the implicit bias of *adversarial training* for DNNs. We believe our results provide a theoretical understanding on the effectiveness of adversarial training for improving robustness against adversarial examples. It could potentially shed light on how to enhance the robustness of adversarially trained models or even further inspire more effective defense mechanisms.

**Organization.** This paper is organized as follows. Section 2 is about notations and settings. Section 3 presents our main results on the implicit bias of adversarial training for DNNs. Section 4 provides numerical experiments to support our claims. We conclude this work in Section 5 and discuss future directions. Some technical proofs are deferred to supplementary materials.

---

**Algorithm 1** Adversarial Training

---

**Input:** Training set $S = \{(x_i, y_i)\}_{i=1}^n$, Adversary $\mathcal{A}$ to solve the inner maximization, learning rate $\eta$, initialization $W_k$ for $k \in \{1, \ldots, L\}$
**for** $t = 0$ **to** $T - 1$ **do**
    $S'(t) = \emptyset$
    **for** $i = 1$ **to** $n$ **do**
        $x_i'(t) = \mathcal{A}(x_i, y_i, W(t))$
        $S'(t) = S'(t) \bigcup (x_i'(t), y_i)$
    **end for**
    **for** $k = 1$ **to** $L$ **do**
        $W_k(t + 1) = W_k(t) - \eta(t)\frac{\partial \tilde{\mathcal{L}}(S'(t);W)}{\partial W_k}$
    **end for**
**end for**

---

## 2 PRELIMINARIES

**Notations.** For any matrix $A \in \mathbb{R}^{m \times n}$, we denote its $i$-th row $j$-th column entry by $A_{ij}$. Let $A^T$ denote the transpose of $A$. $\|\cdot\|_F$ represents the Frobenius norm and $\|\cdot\|_p$ is the $\ell_p$ norm. The training set is $\{(x_i, y_i)\}_{i=1}^n$ where $x_i \in \mathbb{R}^d$, $\|x_i\|_2 \leq 1$ and $y_i \in \{1, -1\}$. For a scalar function $f : \mathbb{R}^d \mapsto \mathbb{R}$, we denote its gradient by $\nabla f$. Furthermore, $\text{tr}(A) = \sum_i A_{ii}$ denotes the trace for the matrix $A$.

We study the adversarial training for the $L$-layer positively homogeneous deep neural network

$$f(x; W) = W_L \varphi_L (W_{L-1} \cdots \varphi_3(W_2\varphi_2(W_1 x)) \cdots) \tag{2}$$

where $W_k$ is the $k$-th layer weight matrix and $\varphi_k$ is the activation function of the $k$-th layer [1]. The multi-$c$-homogeneity of the network is defined by

$$f(x; a_1 W_1, \cdots, a_L W_L) = \prod_{k=1}^{L} a_k^c f(x; W) \tag{3}$$

for any positive constants $a_k$'s and $c \geq 1$, where $W = (W_1, \cdots, W_L)$ is the collection of the parameters of the network. For example, deep ReLU networks are multi-1-homogeneous. For convenience, we also adopt the following notations for a multi-$c$-homogeneous DNN:

$$\rho_k = \|W_k\|_F, \ \rho = \rho_1^c \cdots \rho_L^c \quad \text{and} \quad f(x; W) = \rho f(x; \widehat{W}), \tag{4}$$

where $\widehat{W}_k = W_k / \|W_k\|_F$ with $\|\widehat{W}_k\|_F = 1$ for $k \in \{1, \ldots, L\}$.

We use $\delta_i(W)$ to represent the adversarial perturbation of the original example $x_i$ within the perturbation set $\mathcal{B}(0, \epsilon) : \{\delta : \|\delta\|_p \leq \epsilon\}$ around the original example $x_i$ for $f(x; W)$. Furthermore, we use the *scale invariant adversarial perturbations* defined as follows for adversarial training in this paper.

**Definition 1** (Scale invariant adversarial perturbation). *An adversarial perturbation is said to be a scale invariant adversarial perturbation for $f(x_i; W_1, \ldots, W_L)$ and loss function $\ell$ if it satisfies*

$$\delta_i(a_1 W_1, \ldots, a_L W_L) = \delta_i(W_1, \ldots, W_L) \tag{5}$$

*for any positive constants $a_k$'s.*

We will show in Section 3.2 that FGSM, $\ell_2$-FGM, $\ell_2$-PGD and $\ell_\infty$-PGD perturbations for homogeneous DNNs are all scale invariant perturbations, which are important for analyzing different types of perturbation in a unified manner. The empirical adversarial training loss with the perturbation $\delta_i(W)$ is given by

$$\tilde{\mathcal{L}}(W) = \frac{1}{n} \sum_{i=1}^{n} \tilde{\ell}(y_i, x_i; W) = \frac{1}{n} \sum_{i=1}^{n} \ell\left(y_i f(x_i + \delta_i(W); W)\right). \tag{6}$$

For ease of notation, we denote $\tilde{\ell}_i(W) = \tilde{\ell}(y_i, x_i; W)$. The loss function $\ell$ is continuously differentiable and satisfies

**Assumption 1** (Loss function). $\ell > 0$, $\ell' < 0$, $\lim_{x \to \infty} \ell(x) = 0$ and $\ell_{x \to -\infty}(x) = \infty$.

Many widely used loss functions satisfy the above assumption such as $\ell(x) = e^{-x}$ and the logistic loss $\ell(x) = \ln(1 + e^{-x})$. Furthermore, we make the following common assumptions about the smoothness of $f(x; W)$ and the adversarial perturbation $\delta(W)$:

**Assumption 2** (Smoothness). *With respect to $W$, $yf(x; W)$ is locally Lipschitz for any fixed $x$; $y_i f(x_i; W)$ further have locally Lipschitz gradients and $\delta_i(W)$ are locally Lipschitz for all training examples $x_i$.*

**Remark.** Our results can also be generalized to non-smooth homogeneous neural networks straightforwardly (Appendix B.4). Assuming Lipschitzness about perturbations is because we focus on popular perturbations such as $\ell_2$-FGM and PGD perturbations which have explicit forms and depend on gradients of the network, whose Lipschitzness assumptions are quite common.

## 3 MAIN RESULTS

We present our main theoretical results on the implicit bias of the gradient flow/gradient descent for adversarial training in this section. For the deep linear neural network, a special kind of homogeneous models, in Section 3.1 we further restrict the dataset to be linearly separable and focus on $\ell_2$ adversarial perturbations. We prove that the singular vectors corresponding to the largest singular values of weight matrices get aligned across layers with the progress of adversarial training[2]. Based on this key result, the product of weight matrices converges to the direction of the maximum margin solution

---

[1] $\varphi_k$ is a vector in our notations.
[2] This phenomenon for standard training has been studied by Ji & Telgarsky (2019) first.

under the original data. Furthermore, we study a much more general scenario in Section 3.2: *without the assumption of linearly separability of data and norm constraints on adversarial perturbations*, we show that the gradient flow of adversarial training with scale invariant adversarial perturbations for the homogeneous nonlinear neural networks implicitly performs margin maximization under the *adversarial data*. Due to the space limit, some technical proofs of Section 3.1 are deferred to Appendix A, and we present proofs of Section 3.2 in Appendix B.

## 3.1 ADVERSARIAL TRAINING FOR DEEP LINEAR NETWORK

In this section we restrict the dataset $\{(x_i, y_i)\}_{i=1}^n$ to be linearly separable in the sense that there exists a unit vector $u$ such that $\forall i \in \{1, \ldots, n\}, y_i \langle u, x_i \rangle > 0$ and we explore the adversarial training dynamics of gradient descent for deep linear networks, i.e., with identity activation function $\varphi_k(x) = x$ for all layers,

$$f(x, W) = W_L \cdots W_1 x. \tag{7}$$

Now let

$$\bar{u} = \arg \max_{\|u\|=1} \min_{i \in \{1, \ldots, n\}} y_i \langle x_i, u \rangle$$

be the max-margin solution given by the hard-margin SVM and $\gamma_m = \max_{\|u\|=1} y_m \langle x_m, u \rangle$ be the max-margin, where $m = \arg \min_{i \in \{1, \ldots, n\}} y_i \langle x_i, u \rangle$. We only consider the $\ell_2$-norm adversarial perturbations in this section, i.e., $\mathcal{B}(0, \epsilon) = \{\delta : \|\delta\|_2 \leq \epsilon\}$. The allowed Euclidean distance from an adversarial example $x'$ to its original example $x$ is assumed to satisfy

$$\|x' - x\|_2 \leq \epsilon \leq \gamma_m. \tag{8}$$

We denote the product of all weight matrices by $W_\Pi = W_L \cdots W_1$, which is simply a vector. Our model is adversarially trained by gradient descent

$$W_k(t + 1) = W_k(t) - \eta(t) \frac{\partial \tilde{\mathcal{L}}(t)}{\partial W_k(t)}. \tag{9}$$

The inner maximization of the adversarial training objective Eq. (1) can be solved exactly in this case, where the adversarial perturbations are taken as

$$\delta_i(W(t)) = -y_i \epsilon \frac{W_\Pi(t)}{\|W_\Pi(t)\|_2} \tag{10}$$

for the $t$-th iteration of adversarial training. For any layer $k$, the weight matrix can be decomposed as

$$W_k = U_k \Sigma_k V_k^T$$

where $U_k$ and $V_k$ are the left and right singular matrices, respectively. Furthermore, we use $u_k$ and $v_k$ to represent the left and right singular vectors corresponding to the largest singular value, respectively. We emphasize that the training dynamics abruptly changes due to this perturbation and one should be careful to analyze its effects. It can be easily seen that Eq. (10) is a scale invariant adversarial perturbations by our definition. Note that with this kind of perturbation, the adversarial data during the training are still linearly separable, since we have required that the perturbation is smaller than the max-margin of the original dataset described in Eq. (8). We further assume that $\ell$ satisfies

**Assumption 3.** *The loss function $\ell$ is $\beta$-smooth and $|\ell'| \leq \alpha$.*

such as logistic loss. We are now ready to explore the implicit bias of the gradient descent for adversarial training. Inspired by the idea of studying the smoothness of loss function from Ji & Telgarsky (2019) for standard training, here we first examine whether our adversarial training loss Eq. (1) possesses smoothness. Let the set $\mathcal{S}(r)$ denote the set of weights with bounded norm

$$\mathcal{S}(r) = \left\{ W \big| \|W_k\|_F \leq r, \|W_\Pi\|_2 \geq \bar{r}, k = 1, \ldots, L \right\}, \tag{11}$$

where $r \geq 1$ and $\bar{r}$ is a small constant to avoid trivial solution. To simplify the notation, we let

$$\beta(r, \bar{r}, \epsilon) = r^{3L} L^2 \left[ \alpha + \beta + \frac{\epsilon}{\bar{r}^L} \left( 2\alpha + \beta + \frac{\alpha\beta}{\bar{r}^L} \right) \right]. \tag{12}$$

The following lemma provides us a first view of the overall trends of gradient descent for adversarial training in deep linear network on linear separable dataset.

**Lemma 1** (Overall trends of adversarial training for deep linear network). *With adversarial perturbation Eq. (10), under Assumption 1, Assumption 3 and the requirement Eq. (8), for the deep linear network adversarially trained by gradient descent on linear separable data, the adversarial training objective (1) is $\beta(r, \bar{r}, \epsilon)$-smooth within the given set $\mathcal{S}(r)$ for a constant $r$. However, with constant step size for gradient descent, the weight matrices will eventually leave the set $\mathcal{S}(r)$ if $r$ remains unchanged during the training*

$$\max_{k \in \{1,\ldots,L\}} \|W_k(t_1)\|_F > r \quad \text{for some } t_1 > 0. \tag{13}$$

The first part of this lemma implies the smoothness of the adversarial training loss Eq. (1) while also leading the weight matrices to leave the set $\mathcal{S}(r)$. In fact, this is saying that $\max_{k \in \{1,\ldots,L\}} \|W_k\|_F$ is divergent along the adversarial training trajectory since $\max_{k \in \{1,\ldots,L\}} \|W_k\|_F$ will be larger than $r$ for any given $r$. We can keep $W(t+1)$ inside $\mathcal{S}(r(t))$ at every step $t$ by delicately adjusting the step size of gradient descent and treating $r$ as a function of the step $t$.

**Lemma 2** (Smoothness of the adversarial training loss). *If the learning rate is taken as $\eta(t) = \min\{1, 1/\beta(r, \bar{r}, \epsilon)\}$, then the adversarial training loss $\tilde{\mathcal{L}}(W)$ is $\beta(r, \bar{r}, \epsilon)$-smooth where $r(t+1)$ is taken as*

$$r(t+1) = \begin{cases} r(t) & \text{if } W(t+1) \in \mathcal{S}(r(t) - \mu(t)), \\ r(t) + \mu(t) & \text{otherwise} \end{cases}$$

*during the training, where $\mu = r^{L-1}(1+\epsilon)\alpha/\beta(r, \bar{r}, \epsilon)$.*

We show the divergence of $\max_{k \in \{1,\ldots,L\}} \|W_k\|_F$ along the adversarial training trajectory in Lemma 1, and now we can further derive that the Frobenius norms of weight matrices for all layers are divergent with the smoothness of the adversarial training loss in hand, considering that the differences between any two layers during the adversarial training are always bounded (Lemma 6). Due to this divergence of weight norms, only the direction of the product of weight matrices is important for the model to make predictions.

Now we present our main theorem to provide insights on the theoretical understanding of the implicit bias of gradient descent for adversarial training of deep linear network.

**Theorem 2** (Convergence to the direction of the max-margin solution). *With adversarial perturbation Eq. (10), under Assumption 1, Assumption 3 and requirement (8), for gradient descent of the adversarial training objective Eq. (6) with logistic loss, the singular vectors of the largest singular values of adjacent layers get aligned if the learning rate is taken as that in Lemma 2:*

$$\forall k \in \{1, \ldots, L\} : \quad |\langle u_k, v_{k+1} \rangle| \to 1 \tag{14}$$

*as $t \to \infty$. As a result, the direction of the product of weight matrices converges to that of the max-margin solution*

$$\frac{W_\Pi}{\|W_\Pi\|_2} \to \bar{u}. \tag{15}$$

**Remark.** The alignment phenomenon for standard training has been showed in the previous work Ji & Telgarsky (2019), here we further extend this result to the case of adversarial training, where the training objective is different from that of the standard training due to the introduction of adversarial perturbation dependent on the parameters of the network. Our work is also a significant generalization of Li et al. (2020) which studied the convergence to max-margin solution of adversarial training for logistic regression, a single-layer linear network.

Furthermore, although the direction of the solution Eq. (15) given by adversarial training does not differ from that of standard training, i.e., the direction of $\bar{u}$, these two solutions are not same: the first is maximizing the margin of adversarial examples while the latter is maximizing that of original dataset—which happens to be the same one in this setting. The similarity of these two solutions comes from the fact that the adversarial data are also linearly separable under the requirement Eq. (8), whose max-margin solution has the same direction as that of $\bar{u}$. The above reasoning will be more clear in the next section where we show the solution of adversarial training as a max-margin solution of adversarial examples more explicitly.

## 3.2 Adversarial Training for Homogeneous Deep Neural Network

**Additional definitions** For simplicity, we will use exponential loss $\ell(x) = e^{-x}$ in this section. For an original example $x_i$, the margin for its adversarial example $x_i + \delta_i(W)$ is defined as $\tilde{\gamma}_i = y_i f(x_i + \delta_i(W); W)$ while for the whole dataset $\{(x_i, y_i)\}_{i=1}^n$, the margin for their corresponding adversarial examples is denoted by $\tilde{\gamma}_m$ where $m = \arg\min_{m \in \{1,...,n\}} y_i f(x_i + \delta_i(W); W)$. We introduce the normalized margin as $\hat{\gamma}_i = y_i f(x_i + \delta_i(\widehat{W}); \widehat{W})$ to explore the convergence properties, where $\widehat{W}_k = W_k / \|W_k\|_F$ such that $\|\widehat{W}_k\|_F = 1$ for any layer $k$ . Furthermore, the margin for the original example $x_i$ is defined to be $\gamma_i = y_i f(x_i; W) \geq \tilde{\gamma}_i$. To relieve the headaches of picking suitable learning rates, we consider gradient flow, gradient descent with infinitesimal step, of the training loss Eq. (6), which leads to

$$\frac{dW_k}{dt} = -\left(\frac{\partial \tilde{\mathcal{L}}}{\partial W_k}\right)^T \tag{16}$$

for $k \in \{1, \ldots, L\}$. Note that the introduction of adversarial perturbations (relying on the network parameters) abruptly perturbs the training dynamics. In this section, to conduct a comprehensive study of adversarial training, we do not restrict the perturbation used for solving the inner maximization of the adversarial training objective in advance while only require it to be a scale invariant adversarial perturbation. By noting the following lemma, we can easily generalize our results to the commonly used adversarial attacks:

**Lemma 3.** *Under Assumption 1, $\ell_2$-FGM, FGSM, $\ell_2$-PGD and $\ell_\infty$-PGD perturbations for homogeneous DNNs are all scale invariant adversarial perturbations.*

The RHS of Eq. (16) is different from that of standard training due to the inner maximization. Our key observation is the following property of adversarial training which can drastically simplify our analysis despite of the complexity brought by the adversarial examples.

**Lemma 4.** *For the exponential loss, along the gradient flow trajectory of adversarial training with scale invariant adversarial perturbations for homogeneous DNN $f(x; W)$, we have*

$$tr\left(\frac{\partial \tilde{\mathcal{L}}}{\partial W_k} W_k\right) = -\frac{c}{n} \sum_{i=1}^n e^{-\tilde{\gamma}_i} \tilde{\gamma}_i. \tag{17}$$

In the case of deep linear network, we adopt linearly separable dataset with the requirement Eq. (8), yielding the fact that the adversarial data are also linearly separable. In this section we do not impose such conditions but only apply a weaker assumption in a more general scenario.

**Assumption 4.** *There exists a separability of adversarial examples of the dataset[3]: there exists a time $t_0$ such that $y_i f(x_i + \delta_i(W(t_0)); W(t_0)) > 0$ for all $i \in \{1, \ldots, n\}$.*

Theoretically, Zhang et al. (2020); Gao et al. (2019) proved the convergence to low robust training loss for heavily overparametrized nets. Moreover, adversarial training can typically achieve this separability in practice, i.e., the model can fit adversarial examples of the training dataset, which makes the above assumption a reasonable one. In Section 3.1, we obtain divergent weight norms for adversarial training. Thanks to Lemma 4, we can further generalize it to the case of homogeneous network without linearly separability of dataset and the requirement Eq. (8) for adversarial examples. This divergence of model parameters is necessary for the convergence of the adversarial training loss. To begin with, recalling our definition of $\rho = \prod_{k=1}^L \rho_k^c = \prod_{k=1}^L \|W_k\|_F^c$, we state our first main theorem regarding the overall trend for gradient flow of adversarial training as follows.

**Theorem 3** (Divergence of weight norms). *For the adversarial training objective (6) of the binary classification task with exponential loss and scale invariant adversarial perturbations, under Assumption 2 and Assumption 4, the product of the Frobenius norms of weight matrices diverges to infinity along the trajectory of the gradient flow*

$$\rho = \Omega(\ln t) \to \infty,$$

*and, as a result, the training loss $\tilde{\mathcal{L}}$ converges to zero as $t \to \infty$.*

---

[3]Similar assumptions are previously used in Lyu & Li (2020); Ji & Telgarsky (2020)

**Remark.** The divergence of weight norms for standard training of homogeneous DNNs has been studied by, for example, Lyu & Li (2020); Wei et al. (2019). The above theorem considers *adversarial training*, which implies that the margin for an adversarial example, $\tilde{\gamma}_i$ for any $i \in \{1, \ldots, n\}$, also goes to infinity along the adversarial training trajectory. This makes the normalized margin $\hat{\gamma}_i$ necessary to understand the convergence properties of the adversarial training solution as $t \to \infty$. Built upon this result, we first shed some lights on the implicit bias of the gradient flow for the adversarial training through the following theorem, which discusses the training dynamics of the normalized margins for the adversarial examples.

**Theorem 4** (Non-decreasing adversarial margin). *For adversarial training with scale invariant adversarial perturbations, when the separability of adversarial examples is achieved at $t_0$, the weighted sum of the changing rates of normalized margins for the adversarial examples are non-negative for all $t \geq t_0$:*

$$\forall t \geq t_0 : \quad \sum_{i=1}^n e^{-\rho \hat{\gamma}_i} \frac{d\hat{\gamma}_i}{dt} \geq 0; \tag{18}$$

*there exists a sufficiently large time $t_1 > t_0$ such that $\frac{d\hat{\gamma}_m}{dt} \geq 0$ if $\forall i : \hat{\gamma}_i - \hat{\gamma}_m = \omega(\frac{1}{\rho})$ for all $t \geq t_1$.*

The above theorem reveals a relation for the *normalized margins for adversarial examples*. According to the above lemma, since $\rho \to \infty$ as $t \to \infty$, the term with weight $e^{-\rho \hat{\gamma}_m}$, roughly speaking, dominates the LHS of (18) after some time $t_1 \geq t_0$. Then $d\hat{\gamma}_m/dt$[4] will never be negative for $\forall t \geq t_1$. This directly implies that the gradient flow is actually maximizing the margin for the adversarial examples of the original dataset. Applying Lemma 4 and Theorem 3, in the following theorem, we further strengthen the above observation for adversarial training for homogeneous DNNs.

**Theorem 5** (Convergence to the direction of a KKT point). *Under Assumption 2 and 4, for exponential loss and multi-c-homogeneous DNNs, the limit point of normalized weight parameters $\{W/\|W\|_2 : t \geq 0\}$ of the gradient flow for the adversarial training objective Eq. (6) with scale invariant adversarial perturbations is along the direction of a KKT point of the constrained norm-minimization problem*

$$\min_{W_1, \ldots, W_L} \frac{1}{2} \|W\|_2^2 \quad s.t. \ \tilde{\gamma}_i \geq 1 \quad \forall i \in \{1, \ldots, n\}. \tag{19}$$

**Remark 1.** According to Lemma 3, the above theorem can be directly applied to $\ell_2$-FGM, FGSM, $\ell_2$ and $\ell_\infty$-PGD perturbations. The requirement Eq. (8) in Section 3.1 is not required since the dataset in this section is no longer linearly separable. It is well-known that the norm-minimization problem is closely related to the margin-maximization problem in the sense that

$$\text{Optimization problem}(19) \iff \max_{W_1, \ldots, W_L} \tilde{\gamma}_m, \quad s.t. \ \|W\|_2 = 1.$$

**Remark 2.** Theorem 5 is the key theorem of this section. It shows that, for adversarial training, the gradient flow is implicitly maximizing the margin of *adversarial examples*. Therefore, for perturbations with norm constraints other than $\ell_2$-norm, (19) is no longer an $\ell_2$-norm optimization problem but a mixed-norm one due to the perturbation $\delta(W)$ in $\tilde{\gamma}_m$, which was first observed by Li et al. (2020) for adversarial training of one-layer neural network while our results are for more general deep non-linear homogeneous neural networks. This is one of the major differences between adversarial training and standard training—they are formulated from solving different optimization problem. Theorem 5 provides theoretical understandings of the folklore that adversarial training can modify the decision boundary against adversarial examples to improve robustness.

**Remark 3.** In the context of standard training, Lyu & Li (2020) studied the properties of the normalized margin through its approximation, the smooth-normalized margin. One can try to generalize the analysis method in Lyu & Li (2020) to adversarial training with various kinds of adversarial perturbations by further assuming the local smoothness regarding gradients when considering non-smooth analysis. Generalizinig our results to non-smooth case can be found in Appendix B.4. To easily adapt to adversarial training, our new strategy in this work is to directly analyze the *normalized margin itself for adversarial examples* instead and utilizes the alignment phenomena observed by Ji & Telgarsky (2020). Most importantly, what we seek to push forward in the current work are the theoretical understandings of the adversarial training for DNNs through its implicit bias.

---

[4]Strictly speaking, this is not well-defined since $m$ may not be unique. However, we can use this definition in an approximately correct way for $t$ sufficiently large.

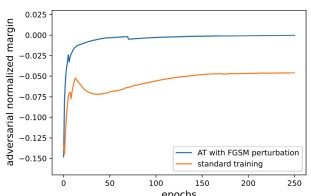 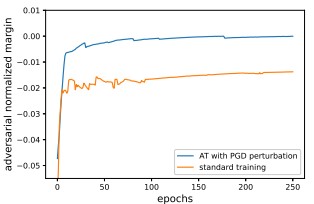 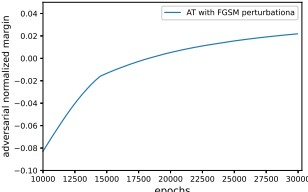

(a) FGSM adversarial examples    (b) $\ell_\infty$-PGD adversarial examples    (c) FGSM adversarial examples

Figure 1: Adversarial training for the 3-layer neural network on MNIST. Adversarial normalized margin for (a) FGSM adversarial examples; (b) $\ell_\infty$-PGD adversarial examples; (c) FGSM adversarial examples after 10000 epochs.

**Trade-off between robustness and accuracy.**   More significantly, Theorem 5 also implies that adversarial training leads to the trade-off between robustness and accuracy as in Zhang et al. (2019). This can be seen as follows. Let $f(\cdot, W)$ denote the classifier obtained from adversarial training. Then for a clean training example $(x, y)$ with $y = 1$ for simplicity, we have $yf(x + \delta(W); W) = f(x + \delta(W); W) > 0$ since $f(x; W)$ is a max-margin classifier over adversarial data. This means that, for any clean test example $(x', y')$ around $x$ in the sense that $x' = x + \tau$ with $\|\tau\|_p \leq \epsilon$, we will have

$$f(x', W) = f(x + \tau; W) \geq f(x + \delta(W); W) > 0 \qquad (20)$$

due to the definition of $\delta(W)$ when $y = 1$: $\delta(W) = \max_{v \in \mathcal{B}(x,\epsilon)} \ell(yf(x + v; W)) = \min_{v \in \mathcal{B}(x,\epsilon)} f(x + v; W)$ for $\ell' < 0$. Therefore, as a result of forcing our model to correctly classify the adversarial example $x + \delta(W)$, it will classify any such $x' = x + \tau$ as having labels 1 even there may be some of them having true labels -1, which leads to the drop of accuracy at the cost of increasing robustness. Besides, since adversarial training maximizes the margin of the specific adversarial perturbation used for training, the trained model performs worse on other perturbations.

## 4    NUMERICAL EXPERIMENTS

In this section, we conduct numerical experiments on MNIST dataset to support our claims. We adversarially trained a 3-layer neural network using SGD with constant learning rate and batch-size 80. The model has the architecture of input layer-$1024$-ReLU-$64$-ReLU-output layer. We present results for adversarial training with: (1) FGSM perturbations with $\epsilon = 16/255$; (2) $\ell_\infty$-PGD perturbations, where the PGD is ran for 5 steps with step size $6/255$ and $\epsilon = 16/255$. As a comparison, we also standardly trained a model with the same architecture to evaluate the normalized margin for adversarial examples by attacking it with FGSM and PGD during its training process.

Fig. 1(a) shows that the normalized margin for FGSM adversarial examples keeps increasing during the process of adversarial training with FGSM perturbations. Meanwhile, the standardly trained model has lower normalized margins for FGSM adversarial examples. Similar results exist for $\ell_\infty$-PGD adversarial examples as showed by Fig. 1(b). To see that adversarial normalized keeps increasing more clearly, we also plot the adversarial training with FGSM perturbations after 10000 epochs in Fig. 1(c). More detailed experiments are in Appendix C. These empirical findings support our claims that adversarial training implicitly maximizes the margin of adversarial examples.

## 5    CONCLUSION

In this paper, we have studied the implicit bias of adversarial training for the deep linear networks and, more generally, the homogeneous DNNs. We proved that adversarial training with scale invariant adversarial perturbations implicitly performs margin-maximization for adversarial data during its training process. Intuitively, such implicit bias strongly implies that adversarial training encourages the neural networks to utilize adversarial examples more to improve its robustness. It will be an interesting future direction to improve the effect of adversarial training by promoting it to enlarge the margins of adversarial examples more effectively. It is also possible to design new defense mechanisms by explicitly requiring the standard training to have the bias of maximizing the margin of adversarial examples. We will leave these potential benefits of our theoretical analysis as future work.

ACKNOWLEDGEMENT

This project is supported by Beijing Nova Program (No. 202072) from Beijing Municipal Science Technology Commission.

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

## A  PROOFS OF SECTION 3.1

In Section A.1, we discuss the smoothness of the adversarial training loss. In Section A.2, we prove the alignment phenomenon of adversarial training for deep linear networks. We make the common assumption that $\tilde{\mathcal{L}}(W(0)) \leq \tilde{\mathcal{L}}(0) = \ell(0)$ and $\nabla \tilde{\mathcal{L}}(W(0)) \neq 0$ to start adversarial training.

## A.1 SMOOTHNESS OF THE ADVERSARIAL TRAINING LOSS

### A.1.1 PROOF OF LEMMA 1

*Proof.* We first prove that the adversarial training loss for the deep linear network is smooth within $\mathcal{S}(r)$. For any $W, V \in \mathcal{S}(r)$, let $\delta_W$ and $\delta_V$ denote the adversarial perturbations of the network $f(x; W)$ and $f(x; V)$, respectively. Then we have, by taking derivatives of $\mathcal{L}$ w.r.t $W$ and using that $\|A\|_F + \|B\|_F \geq \|A - B\|_F$,

$$\left\| \frac{\partial \tilde{\mathcal{L}}}{\partial W_1} - \frac{\partial \tilde{\mathcal{L}}}{\partial V_1} \right\|_F \leq \underbrace{\frac{1}{n} \sum_{i=1}^{n} \left\| W_2^T \cdots W_L^T x_i^T y_i \tilde{\ell}_i'(W_\Pi) - V_2^T \cdots V_L^T x_i^T y_i \tilde{\ell}_i'(V_\Pi) \right\|_F}_{(a)}$$

$$+ \underbrace{\frac{\epsilon}{n} \sum_{i=1}^{n} \left\| \frac{\partial \|W_\Pi\|}{\partial W_1} \tilde{\ell}_i'(W_\Pi) - \frac{\partial \|V_\Pi\|}{\partial V_1} \tilde{\ell}_i'(V_\Pi) \right\|_F}_{(b)}, \tag{21}$$

where the term (b) abruptly changes the training dynamics. For simplicity, we use $x, y$ and $\tilde{\ell}'$ without the subscript $i$ because the analysis can be done similarly for any $i \in \{1, \ldots, n\}$. We begin with the term (a) of Eq. (21):

$$\begin{aligned}
(a) &\leq \left\| W_2^T \cdots W_L^T x^T y \tilde{\ell}'(W_\Pi) - V_2^T W_3^T \cdots W_L^T x^T y \tilde{\ell}'(W_\Pi) \right\| \\
&\quad + \left\| V_2^T W_3^T \cdots W_L^T x^T y \tilde{\ell}'(W_\Pi) - V_2^T \cdots V_L^T x^T y \tilde{\ell}'(V_\Pi) \right\| \\
&\leq \|W_2 - V_2\| \left\| W_3^T \cdots W_L^T x^T y \tilde{\ell}'(W_\Pi) \right\| \\
&\quad + \|V_2\| \left\| W_3^T \cdots W_L^T x^T y \tilde{\ell}'(W_\Pi) - V_3^T \cdots V_L^T x^T y \tilde{\ell}'(V_\Pi) \right\| \\
&\overset{c}{\leq} \|W_2 - V_2\| r^{L-2} \alpha + r \left\| W_3^T \cdots W_L^T x^T y \tilde{\ell}'(W_\Pi) - V_3^T \cdots V_L^T x^T y \tilde{\ell}'(V_\Pi) \right\| \\
&\overset{d}{\leq} \sum_{k=2}^{L} r^{L-2} \alpha \|W_k - V_k\| + r^{L-2} \|V_L\| \left\| \tilde{\ell}'(W_\Pi) - \tilde{\ell}'(V_\Pi) \right\| \\
&\overset{e}{\leq} \left( r^{L-2} \alpha (L-1) + \beta L r^{2L-2} \right) \|W - V\|, \tag{22}
\end{aligned}$$

where c follows from $\|x\| \leq 1$, $\|W_k\| \leq r$ and $\ell' \leq \alpha$; repeating the procedure of c until $k = L$ gives us d; e is because $\|W_k - V_k\| \leq \|W - V\|$ for any $k$ and

$$\left\| \tilde{\ell}'(W_\Pi) - \tilde{\ell}'(V_\Pi) \right\| \leq \beta \|W_\Pi - V_\Pi\| \leq \beta L r^{L-1} \|W - V\| \tag{23}$$

considering that $\ell$ is a $\beta$-smooth function. It now remains us to bound the term (b) of Eq. (21). Note that

$$\frac{\partial \|W_\Pi\|_2}{\partial W_1} = \frac{W_2^T \cdots W_L^T W_L \cdots W_1}{\|W_\Pi\|_2}, \tag{24}$$

then we have

$$\text{(b)} \leq \epsilon \left\| \frac{W_2^T \cdots W_L^T W_L \cdots W_1}{\|W_\Pi\|} \tilde{\ell}'(W_\Pi) - \frac{V_2^T W_3^T \cdots W_L^T W_L \cdots W_1}{\|W_\Pi\|} \tilde{\ell}'(W_\Pi) \right\|$$

$$+ \epsilon \left\| \frac{V_2^T W_3^T \cdots W_L^T W_L \cdots W_1}{\|W_\Pi\|} \tilde{\ell}'(W_\Pi) - \frac{V_2^T V_3^T \cdots V_L^T V_L \cdots V_1}{\|V_\Pi\|} \tilde{\ell}'(V_\Pi) \right\|$$

$$\overset{\text{c}}{\leq} \frac{\epsilon \alpha r^{2L-2}}{\bar{r}^L} \|W_2 - V_2\|$$

$$+ \epsilon r \left\| \frac{W_3^T \cdots W_L^T W_L \cdots W_1}{\|W_\Pi\|} \tilde{\ell}'(W_\Pi) - \frac{V_3^T \cdots V_L^T V_L \cdots V_1}{\|V_\Pi\|} \tilde{\ell}'(V_\Pi) \right\|$$

$$\overset{\text{d}}{\leq} (2L-1) \frac{\epsilon \alpha r^{2L-2}}{\bar{r}^L} \|W - V\| + \epsilon r^{2L-2} \left\| \frac{\tilde{\ell}'(W_\Pi)}{\|W_\Pi\|} - \frac{\tilde{\ell}'(V_\Pi)}{\|V_\Pi\|} \right\|$$

$$\overset{\text{e}}{\leq} (2L-1) \frac{\epsilon \alpha r^{2L-2}}{\bar{r}^L} \|W - V\| + \frac{\epsilon r^{2L-2}}{\bar{r}^L} \left\| \tilde{\ell}'(W_\Pi) - \tilde{\ell}'(V_\Pi) \right\| + \frac{\epsilon \alpha r^{2L-2}}{\bar{r}^{2L}} \|W_\Pi - V_\Pi\|$$

$$\overset{\text{f}}{\leq} \frac{\epsilon r^{2L-2}}{\bar{r}^L} \left( (2L-1)\alpha + \beta L r^{L-1} + \frac{\alpha \beta L r^{L-1}}{\bar{r}^L} \right) \|W - V\|, \tag{25}$$

where c follows from $\|W_k\| \leq r$ and $\tilde{\ell}' \leq \alpha$; repeating c for $2L-1$ times gives d; e follows from $\|\|W_\Pi\| - \|V_\Pi\|\| \leq \|W_\Pi - V_\Pi\|$; invoking Eq. (23) gives us f. Now we can bound Eq. (21) by combing the above results

$$\left\| \frac{\partial \tilde{\mathcal{L}}}{\partial W_1} - \frac{\partial \tilde{\mathcal{L}}}{\partial V_1} \right\|_F$$

$$\leq \left[ r^{L-2}\alpha(L-1) + \beta r^{2L-2}L + \frac{\epsilon r^{2L-2}}{\bar{r}^L} \left( (2L-1)\alpha + \beta L r^{L-1} + \frac{\alpha \beta L r^{L-1}}{\bar{r}^L} \right) \right] \|W - V\|$$

$$\leq r^{3L} L \left[ \alpha + \beta + \frac{\epsilon}{\bar{r}^L} \left( 2\alpha + \beta + \frac{\alpha \beta}{\bar{r}^L} \right) \right] \|W - V\|, \tag{26}$$

where the last inequality is because we choose $r \geq 1$. The above analysis is for the first layer. Now we can apply the same technique to other layers, which finally gives us

$$\|\nabla \tilde{\mathcal{L}}(W) - \nabla \tilde{\mathcal{L}}(V)\| \leq r^{3L} L^2 \left[ \alpha + \beta + \frac{\epsilon}{\bar{r}^L} \left( 2\alpha + \beta + \frac{\alpha \beta}{\bar{r}^L} \right) \right] \|W - V\|, \tag{27}$$

thus $\tilde{\mathcal{L}}$ is $\beta(r, \bar{r}, \epsilon)$-smooth inside $\mathcal{S}(r)$ where

$$\beta(r, \bar{r}, \epsilon) = r^{3L} L^2 \left[ \alpha + \beta + \frac{\epsilon}{\bar{r}^L} \left( 2\alpha + \beta + \frac{\alpha \beta}{\bar{r}^L} \right) \right]. \tag{28}$$

We now prove the second statement of Lemma 1, which says that $W$ will leave $\mathcal{S}(r)$ with constant $r$ and learning rate $\eta$. Let $W(t), W(t+1) \in \mathcal{S}(r)$ and $\eta(t) = \min\{1, 1/\beta(r, \bar{r}, \epsilon)\}$.

**Lemma 5** (Adapted from Lemma 6 in Ji & Telgarsky (2019)). *Under Assumption 1 and Assumption 3, suppose gradient descent for adversarial training is run with a constant step size $1/\beta(r, \epsilon, \bar{r})$. Then there exists a time $t$ when $\max_k \|W_k(t)\|_F > r$.*

Eq. (27) states that $\tilde{\mathcal{L}}$ is $\beta(r, \bar{r}, \epsilon)$-smooth, therefore

$$\tilde{\mathcal{L}}(W(t+1)) - \tilde{\mathcal{L}}(W(t)) \leq \left\langle \nabla \tilde{\mathcal{L}}(W(t)), -\eta(t)\tilde{\mathcal{L}}(W(t)) \right\rangle + \frac{\beta(r, \bar{r}, \epsilon)\eta(t)^2}{2} \|\nabla \tilde{\mathcal{L}}(W(t))\|^2$$

$$= -\frac{\eta(t)}{2} \|\nabla \tilde{\mathcal{L}}(W(t))\|^2, \tag{29}$$

which also implies that $\tilde{\mathcal{L}}(W(t))$ never increases during the training

$$\tilde{\mathcal{L}}(W(t+1)) \leq \tilde{\mathcal{L}}(W(t)) - \frac{\eta(t)}{2} \|\nabla \tilde{\mathcal{L}}(W(t))\|^2. \tag{30}$$

By repeatedly applying Lemma 5 when the learning rate is constant, we know $W$ will eventually leave $\mathcal{S}(r)$, which also implies that $\max_k \|W_k\|_F$ is unbounded in this case. $\qquad \square$

### A.1.2 PROOF OF LEMMA 2

*Proof.* For any step $t$ with the learning rate in the Lemma 2, suppose that $W(t) \in \mathcal{S}(r(t) - \mu(t))$, we then have

$$\|W_k(t+1)\|_F \leq \|W_k(t)\|_F + \eta(t) \left\| \frac{\partial \tilde{\mathcal{L}}}{\partial W_k(t)} \right\|_F \leq \|W_k(t)\|_F + \mu(t),$$

where the last inequality follows from

$$\left\| \frac{\partial \tilde{\mathcal{L}}}{\partial W_k(t)} \right\| \leq \left\| W_{k+1} \cdots W_L^T x^T W_1^T \cdots W_{k-1}^T y \tilde{\ell}'(W_\Pi) \right\| + \epsilon \left\| \frac{\partial \|W_\Pi\|_2}{\partial W_k} \tilde{\ell}'(W_\Pi) \right\|$$

$$\leq (1 + \frac{\epsilon}{r^L}) \alpha r^{2L}. \tag{31}$$

Therefore $\|W_k(t+1)\|_F$ will always stay in $\mathcal{S}(r(t))$ and the smoothness is preserved. If $\|W_k(t+1)\|_F$ is still less than $r(t) - \mu(t)$, then we keep $r(t+1) = r(t)$ while we increase $r(t+1)$ to

$$r(t+1) = r(t) + \mu(t)$$

if $r(t) - \mu(t) \leq \|W_k(t+1)\|_F \leq r(t)$ such that $\|W_k(t+1)\|_F \leq r(t) \leq r(t+1) - \mu(t+1)$, which means that $W(t+2)$ will stay inside $\mathcal{S}(r(t+1))$. The overall smoothness can then be proved by induction. $\qquad\square$

### A.2 ALIGNMENT PHENOMENON

We start with the following lemma for the adversarial training to prove the alignment phenomenon[5]:

**Lemma 6** (Bounded differences between weight norms). *The difference of Frobenius norms for any two layers is bounded during the adversarial training process*

$$\left| \|W_k(t)\|_F^2 - \|W_j(t)\|_F^2 \right| - 2\tilde{\mathcal{L}}(W(0)) \leq \left| \|W_k(0)\|_F^2 - \|W_j(0)\|_F^2 \right|, \ \forall k,j \in \{1,\ldots,L\}.$$

*Proof.* If there is not adversarial perturbation, then we can easily obtain

$$W_{k+1}^T \frac{\partial \mathcal{L}}{\partial W_{k+1}} = \frac{\partial \mathcal{L}}{\partial W_k} W_k^T \tag{32}$$

by conducting some algebra, where $\mathcal{L}$ stands for the empirical loss without adversarial perturbation. Then invoking Eq. (24) for $k$ by noting that $\operatorname{tr}(A) = \operatorname{tr}(A^T)$ and $\operatorname{tr}(ABC) = \operatorname{tr}(CAB)$, we have

$$y\epsilon\tilde{\ell}(W_\Pi) W_{k+1}^T \frac{\partial \|W_\Pi\|_2}{\partial W_{k+1}} = \frac{y\epsilon\tilde{\ell}(W_\Pi)}{\|W_\Pi\|_2} W_{k+1}^T \cdots W_L^T W_L \cdots W_1 W_1^T \cdots W_k^T$$

$$= y\epsilon\tilde{\ell}(W_\Pi) \frac{\partial \|W_\Pi\|_2}{\partial W_k} W_k^T. \tag{33}$$

Combining Eq. (32), for the adversarial empirical loss, we have

$$W_{k+1}^T \frac{\partial \tilde{\mathcal{L}}}{\partial W_{k+1}} = \frac{\partial \tilde{\mathcal{L}}}{\partial W_k} W_k^T, \tag{34}$$

which will give us the following relation

$$W_{k+1}^T(t+1)W_{k+1}(t+1) = W_{k+1}^T(t)W_{k+1}(t) + \eta(t)^2 \left( \frac{\partial \tilde{\mathcal{L}}}{\partial W_{k+1}(t)} \right)^T \left( \frac{\partial \tilde{\mathcal{L}}}{\partial W_{k+1}(t)} \right)$$

$$- \eta(t) \left[ W_{k+1}^T(t) \left( \frac{\partial \tilde{\mathcal{L}}}{\partial W_{k+1}(t)} \right) + \left( \frac{\partial \tilde{\mathcal{L}}}{\partial W_{k+1}(t)} \right)^T W_{k+1}(t) \right],$$

---

[5]This property for standard training has previously been studied by Ji & Telgarsky (2019); Arora et al. (2018).

which can be easily seen by writing out the gradient descent update explicitly. One can derive a similar relation for $W_k(t+1)W_k^T(t+1)$. Combing Eq. (32), we have

$$W_{k+1}^T(t+1)W_{k+1}(t+1) - W_{k+1}^T(t)W_{k+1}(t) - \eta(t)^2 \left(\frac{\partial\tilde{\mathcal{L}}}{\partial W_{k+1}(t)}\right)^T \left(\frac{\partial\tilde{\mathcal{L}}}{\partial W_{k+1}(t)}\right)$$

$$= W_k(t+1)W_k^T(t+1) - W_k(t)W_k^T(t) - \eta(t)^2 \left(\frac{\partial\tilde{\mathcal{L}}}{\partial W_k(t)}\right) \left(\frac{\partial\tilde{\mathcal{L}}}{\partial W_k(t)}\right)^T. \tag{35}$$

Summing up the above equation from $0$ to $t$ will give us

$$W_{k+1}^T(t)W_{k+1}(t) - W_{k+1}^T(0)W_{k+1}(0) - A_{k+1}(t) = W_k(t)W_k^T(t) - W_k(0)W_k^T(0) - B_k(t) \tag{36}$$

where symmetric matrices $A_k$ and $B_k$ are defined by

$$A_k(t) = \sum_{\tau=0}^{t-1} \eta(\tau)^2 \left(\frac{\partial\tilde{\mathcal{L}}}{\partial W_k(\tau)}\right)^T \left(\frac{\partial\tilde{\mathcal{L}}}{\partial W_k(\tau)}\right)$$

and

$$B_k(t) = \sum_{\tau=0}^{t-1} \eta(\tau)^2 \left(\frac{\partial\tilde{\mathcal{L}}}{\partial W_k(\tau)}\right) \left(\frac{\partial\tilde{\mathcal{L}}}{\partial W_k(\tau)}\right)^T.$$

Also notice that

$$\mathrm{tr}\left(A_k(t)\right) = \mathrm{tr}\left(B_k(t)\right) \tag{37}$$

where

$$\mathrm{tr}\left(A_k(t)\right) \leq \sum_{k'=1}^{L} \mathrm{tr}\left(\sum_{\tau=0}^{t-1} \eta(\tau)^2 \left(\frac{\partial\tilde{\mathcal{L}}}{\partial W_{k'}(\tau)}\right)^T \left(\frac{\partial\tilde{\mathcal{L}}}{\partial W_{k'}(\tau)}\right)\right)$$

$$= \sum_{k'=1}^{L}\sum_{\tau=0}^{t-1} \eta(\tau)^2 \left\|\frac{\partial\tilde{\mathcal{L}}}{\partial W_{k'}(\tau)}\right\|_F^2$$

$$\overset{\mathsf{a}}{\leq} \sum_{\tau=0}^{t-1} \eta(\tau) \left\|\nabla\tilde{\mathcal{L}}(W(\tau))\right\|^2$$

$$\overset{\mathsf{b}}{\leq} 2\sum_{\tau=0}^{t-1} \eta(\tau) \left[\tilde{\mathcal{L}}(W(\tau)) - \tilde{\mathcal{L}}(W(\tau+1))\right]$$

$$\overset{\mathsf{c}}{=} 2\tilde{\mathcal{L}}(W(0)) - 2\tilde{\mathcal{L}}(W(t)) \leq 2\tilde{\mathcal{L}}(W(0)), \tag{38}$$

where $\mathsf{a}$ is because $\eta < 1$; $\mathsf{b}$ is due to Eq. (29); summing up from $\tau = 0$ to $t-1$ and noting that $\tilde{\mathcal{L}} > 0$ give us $\mathsf{c}$. For any two layers $k < j$, taking trace of both sides of (36) and summing from $k$ to $j$, we have that the differences between the Frobenius norms for any two layers

$$\left|\|W_k(t)\|_F^2 - \|W_j(t)\|_F^2\right| = \left|\mathrm{tr}\left(A_j(t)\right) + \|W_j(0)\|_F^2 - \mathrm{tr}\left(B_k(t)\right) - \|W_k(0)\|_F^2\right|$$

$$\leq 2\tilde{\mathcal{L}}(W(0)) + \left|\|W_j(0)\|_F^2 - \|W_k(0)\|_F^2\right|$$

are always bounded. Since $\max_{1\leq k\leq L} \|W_k\|_F \to \infty$, the above result implies that $\|W_k\|_F \to \infty$ for all $k$. $\qquad\square$

### A.2.1 PROOF OF THEOREM 2

Intuitively, $W_k$ can be decomposed as

$$W_k(t) = U_k(t)\Sigma_k(t)V_k^T(t).$$

For $W_k$, denote the first singular value, the corresponding left singular vector and right singular vector by $\sigma_k, u_k$ and $v_k$. Since $\sigma_k \to \infty$, the initialization becomes negligible as $t \to \infty$. According to Eq. (36), this will lead to

$$U_k(t)\Sigma_k(t)\Sigma_k^T(t)U_k^T(t) \to V_{k+1}(t)\Sigma_{k+1}^T(t)\Sigma_{k+1}(t)V_{k+1}^T(t)$$

as $t \to \infty$. On the other hand, since $W_L W_L^T$ has rank 1 for $W_L$ being a row vector, all layers have rank 1 because they get aligned with $W_L$ and $\frac{W_k}{\|W_k\|_F} \to u_k v_k^T$.

To further elaborate the above reasoning, we provide a more exact proof starting with the definition of the alignment phenomenon.

**Definition 2** (Alignment phenomenon for deep linear networks.)**.** *For deep linear network $f(x; W) = W_L \cdots W_1 x$, the alignment phenomenon is defined as*

$$\forall k \in \{1, \ldots, k\} : |\langle u_k, v_{k+1} \rangle| \to 1 \tag{39}$$

*and $W_k/\|W_k\|_F \to u_k v_k^T$ as $t \to \infty$ along the training trajectory, where $u_k, v_k$ are the left and right singular vectors correspond to the largest singular value of $W_k$.*

The proof for the alignment phenomenon of adversarially trained deep linear networks in our setting is as follows [6].

*Proof.* Note that

$$\langle u_k, v_{k+1} \rangle^2 \sigma_{k+1}^2 = u_k^T W_{k+1}^T W_{k+1} u_k + u_k^T (v_{k+1} \sigma_{k+1}^2 v_{k+1}^T - W_{k+1}^T W_{k+1}) u_k, \tag{40}$$

we have

$$\frac{u_k^T W_{k+1}^T W_{k+1} u_k}{\sigma_{k+1}^2} + \frac{\|W_{k+1}\|_2^2 - \|W_{k+1}\|_F^2}{\sigma_{k+1}^2} \le \langle u_k, v_{k+1} \rangle^2 \le 1 \tag{41}$$

by utilizing the definition of matrix norm and

$$\|W_{k+1}\|_F^2 \ge u_k^T W_{k+1}^T W_{k+1} u_k. \tag{42}$$

According to Eq. (36), let

$$\Gamma_k = W_{k+1}^T(0)W_{k+1}(0) - W_k(0)W_k(0)^T + A_{k+1} - B_k, \tag{43}$$

and replace $W_{k+1}^T W_{k+1}$ with $\Gamma_k + W_k W_k^T$ in Eq. (41), we have

$$\langle u_k, v_{k+1} \rangle^2 \ge \frac{\sigma_k^2}{\sigma_{k+1}^2} + \frac{u_k^T \Gamma_k u_k + \|W_{k+1}\|_2^2 - \|W_{k+1}\|_F^2}{\sigma_{k+1}^2}. \tag{44}$$

To prove the alignment phenomenon, we now bound the RHS of Eq. (44) with $1 - \alpha$ where $\alpha \to 0$ when $t \to \infty$. We are going to bound 3 terms, namely $u_k^T \Gamma_k u_k$, $\|W_k\|_2^2 - \|W_k\|_F^2$ and $\sigma_k^2/\sigma_{k+1}^2$.

1. Bound $u_k^T \Gamma_k u_k$.

$$\begin{aligned} u_k^T \Gamma_k u_k &\ge -u_k^T W_k(0) W_k(0)^T u_k - u_k^T B_k u_k \\ &\ge -\|W_k(0)\|_2^2 - \text{tr}(B_k) \\ &\ge -\|W_k(0)\|_2^2 - 2\tilde{\mathcal{L}}(W(0)), \end{aligned} \tag{45}$$

where we used Eq. (38) in the third inequality.

2. Bound $\|W_k\|_2^2 - \|W_k\|_F^2$. According to the definition of singular values, we have

$$\begin{aligned} \|W_k\|_2^2 = \sigma_k^2 &\ge v_{k+1}^T W_k W_k^T v_{k+1} \\ &= v_{k+1}^T (W_{k+1}^T W_{k+1} - \Gamma_k) v_{k+1} \\ &\ge \sigma_{k+1}^2 - v_{k+1}^T \Delta W_k(0) v_{k+1} - \text{tr}(B_k) \end{aligned} \tag{46}$$

---

[6]Some parts of the proof is inspired by Ji & Telgarsky (2019)

where $\Delta W_k(0) = W_k(0)W_k(0)^T - W_{k+1}^T(0)W_{k+1}(0)$ and we used that $\text{tr}\,(B_k) \geq \|B_k\|_2$ in the second inequality. On the other hand, by taking trace of both sides of Eq. (36), we have

$$\|W_k\|_F^2 = \|W_{k+1}\|_F^2 - \|W_{k+1}(0)\|_F^2 + \|W_k(0)\|_F^2 - \text{tr}\,(A_{k+1}) + \text{tr}\,(B_k). \quad (47)$$

Combined with Eq. (46), we have

$$\begin{aligned}
\|W_k\|_2^2 - \|W_k\|_F^2 \geq\; & \|W_{k+1}\|_2^2 - \|W_{k+1}\|_F^2 \\
& + (\|W_{k+1}(0)\|_F^2 - \|W_k(0)\|_F^2) - v_{k+1}^T \Delta W(0) v_{k+1} \\
& + (\text{tr}\,(A_{k+1}) - \text{tr}\,(B_k)) - \text{tr}\,(B_k) \\
\overset{a}{\geq}\; & \|W_{k+2}\|_2^2 - \|W_{k+2}\|_F^2 \\
& + (\|W_{k+2}(0)\|_F^2 - \|W_k(0)\|_F^2) - (v_{k+1}^T \Delta W_k(0) v_{k+1} + v_{k+2}^T \Delta W_{k+1}(0) v_{k+2}) \\
& + (\text{tr}\,(A_{k+2}) - \text{tr}\,(B_k)) - (\text{tr}\,(B_k) + \text{tr}\,(B_{k+1})) \\
& \cdots \\
\overset{b}{\geq}\; & \|W_L(0)\|_F^2 - \|W_k(0)\|_F^2 - \sum_{k'=k}^{L-1} v_{k'+1}^T \Delta W_{k'}(0) v_{k'+1} - \sum_{k'=k}^{L-1} \text{tr}\,(B_{k'}) \\
& + \text{tr}\,(A_L) - \text{tr}\,(B_k) \\
\overset{c}{\geq}\; & M - 2L\tilde{\mathcal{L}}(W(0)), \quad (48)
\end{aligned}$$

where a follows from Eq. (37); b follows from summing a from $k$ to $L-1$ and that $\|W_L\|_2 = \|W_L\|_F$; c is because Eq. (38) and that

$$M = \min \|W_L(0)\|_F^2 - \|W_k(0)\|_F^2 - \sum_{k'=k}^{L-1} v_{k'+1}^T \Delta W_{k'}(0) v_{k'+1} \quad (49)$$

is finite.

3. Bound $\sigma_k^2/\sigma_{k+1}^2$. Eq. (46) gives us that

$$\frac{\sigma_k^2}{\sigma_{k+1}^2} \geq 1 - \frac{v_{k+1}^T \Delta W_k(0) v_{k+1} - \text{tr}\,(B_k)}{\sigma_{k+1}^2} \geq 1 - \frac{v_{k+1}^T \Delta W_k(0) v_{k+1} + 2\tilde{\mathcal{L}}(W(0))}{\sigma_{k+1}^2}, \quad (50)$$

where $v_{k+1}^T \Delta W_k(0) v_{k+1} + 2\tilde{\mathcal{L}}(W(0))$ is finite.

Putting the bounds obtained from the above 1, 2 and 3 back to Eq. (44) will give us

$$\langle u_k, v_{k+1} \rangle^2 \geq 1 - \frac{\|W_k(0)\|_2^2 + 2(L+2)\tilde{\mathcal{L}}(W(0)) - M + v_{k+1}^T \Delta W_k(0) v_{k+1}}{\sigma_{k+1}^2}. \quad (51)$$

Considering that $\sigma_k \to \infty$ and the numerator is finite, we thus conclude that the RHS of Eq. (51) will converge to 1. Therefore, we have that

$$\langle u_k, v_{k+1} \rangle^2 \to 1 \quad (52)$$

as $t \to \infty$. Furthermore, since $\|W_{k+1}\|_F^2 - \|W_{k+1}\|_2^2$ is finite (see Eq. (48)) and $\sigma_k \to \infty$ (i.e., $\|W_k\|_2^2/\|W_k\|_F^2 \to 1$), we have $\frac{W_k}{\|W_k\|_F} \to u_k v_k^T$ because other singular values are negligible. We now have the alignment phenomenon. $\qquad \square$

Furthermore, we have

$$\frac{W_\Pi}{\|W_L\|_F \cdots \|W_1\|_F} \to v_1 u_1^T \cdots v_L u_L^T \to v_1. \quad (53)$$

We now move forward to prove the alignment phenomenon for the first layer. We assume that the support vectors span the data space $\mathbb{R}^d$ and denote the orthogonal complement of $\text{span}(\bar{u})$ by $\bar{u}^\perp$. Let $\mathcal{P}_\perp$ denote the projection onto $\bar{u}^\perp$. Under this assumption, Ji & Telgarsky (2019) showed the following lemma[7]

---

[7]This lemma is based on Lemma 12 in Soudry et al. (2018).

**Lemma 7** (Lemma 3 in Ji & Telgarsky (2019)). *Let $S$ denote the set of indices of support vectors, then with probability 1,*

$$\kappa := \min_{|\xi|=1, \xi \perp \bar{u}} \max_{i \in S} \langle \xi, x_i y_i \rangle > 0 \tag{54}$$

*if the data is sampled from absolutely continuous distribution.*

To begin with, we first introduce the following lemma.

**Lemma 8.** *For the logistic loss function $\ell = \ln(1 + e^{-x})$, if $\langle W_\Pi, \bar{u} \rangle \geq 0$ and $\|\mathcal{P}_\perp W_\Pi\| \geq 2n/e\kappa = \varphi$, then*

$$\left\langle \mathcal{P}_\perp W_1, \frac{\partial \tilde{\mathcal{L}}}{\partial W_1} \right\rangle \geq 0. \tag{55}$$

*Proof.* We first decompose Eq. (55) as two parts:

$$\left\langle \mathcal{P}_\perp W_1, \frac{\partial \tilde{\mathcal{L}}}{\partial W_1} \right\rangle = \frac{1}{n} \sum_{i=1}^{n} \tilde{\ell}'_i \left[ y_i \left\langle \mathcal{P}_\perp W_1, \frac{\partial f(x_i; W)}{\partial W_1} \right\rangle - \epsilon \left\langle \mathcal{P}_\perp W_1, \frac{\partial \|W_\Pi\|}{\partial W_1} \right\rangle \right], \tag{56}$$

where

$$\left\langle \mathcal{P}_\perp W_1, \frac{\partial \|W_\Pi\|}{\partial W_1} \right\rangle = \frac{1}{\|W_\Pi\|} \left\langle \mathcal{P}_\perp W_1, W_1^T \cdots W_L^T W_L \cdots W_2 \right\rangle$$

$$= \frac{1}{\|W_\Pi\|} \left\langle \mathcal{P}_\perp W_\Pi, \mathcal{P}_\perp W_\Pi \right\rangle \geq 0.$$

Since $\tilde{\ell}' < 0$, we have

$$\left\langle \mathcal{P}_\perp W_1, \frac{\partial \tilde{\mathcal{L}}}{\partial W_1} \right\rangle \geq \frac{1}{n} \sum_{i=1}^{n} \tilde{\ell}'_i y_i \left\langle \mathcal{P}_\perp W_1, \frac{\partial f(x_i; W)}{\partial W_1} \right\rangle = \frac{1}{n} \sum_{i=1}^{n} \left\langle \mathcal{P}_\perp W_\Pi, x_i y_i \tilde{\ell}'(W_\Pi) \right\rangle. \tag{57}$$

Let $x_j y_j \in \arg\max_{i \in S} \langle -\mathcal{P}_\perp W_\Pi, x_i y_i \rangle$ which means that

$$\langle -\mathcal{P}_\perp W_\Pi, x_j y_j \rangle \geq \kappa \|\mathcal{P}_\perp W_\Pi\|.$$

Then we have

$$\frac{1}{n} \sum_{i=1}^{n} \left\langle \mathcal{P}_\perp W_\Pi, x_i y_i \tilde{\ell}'(W_\Pi) \right\rangle$$

$$= \frac{1}{n} \sum_{i=1}^{n} \frac{e^{-\langle W_\Pi, (x_i + \delta_i) y_i \rangle}}{1 + e^{-\langle W_\Pi, (x_i + \delta_i) y_i \rangle}} \left\langle -\mathcal{P}_\perp W_\Pi, x_i y_i \right\rangle$$

$$= \frac{1}{n} \sum_{i=1}^{n} \frac{e^{-\langle W_\Pi, (x_i + \delta_i) y_i \rangle}}{1 + e^{-\langle W_\Pi, (x_i + \delta_i) y_i \rangle}} \left\langle -\mathcal{P}_\perp W_\Pi, \mathcal{P}_\perp x_i y_i \right\rangle$$

$$\geq \frac{1}{n} \sum_{i=1}^{n} \frac{e^{-\langle W_\Pi, (x_i + \delta_i) y_i \rangle}}{1 + e^{-\langle W_\Pi, (x_i + \delta_i) y_i \rangle}} \left\langle -\mathcal{P}_\perp W_\Pi, \mathcal{P}_\perp x_i y_i \right\rangle$$

$$\geq \frac{1}{n} \frac{e^{-\langle W_\Pi, (x_j + \delta_j) y_j \rangle}}{1 + e^{-\langle W_\Pi, (x_j + \delta_j) y_j \rangle}} \left\langle -\mathcal{P}_\perp W_\Pi, \mathcal{P}_\perp x_i y_i \right\rangle + \frac{1}{n} \sum_{k \in C_k} \frac{e^{-\langle W_\Pi, (x_k + \delta_k) y_k \rangle}}{1 + e^{-\langle W_\Pi, (x_k + \delta_k) y_k \rangle}} \left\langle -\mathcal{P}_\perp W_\Pi, \mathcal{P}_\perp x_k y_k \right\rangle, \tag{58}$$

where $C_k = \{k : \langle -\mathcal{P}_\perp W_\Pi, \mathcal{P}_\perp x_k y_k \rangle \leq 0\}$. Considering that $x_j y_j$ is a support vector, the first term of (58) can be bounded as follows:

$$\frac{1}{n} \frac{e^{-\langle W_\Pi, (x_j + \delta_j) y_j \rangle}}{1 + e^{-\langle W_\Pi, (x_j + \delta_j) y_j \rangle}} \left\langle -\mathcal{P}_\perp W_\Pi, \mathcal{P}_\perp x_i y_i \right\rangle$$

$$\geq \frac{\kappa}{n} \|\mathcal{P}_\perp W_\Pi\| e^{-\langle W_\Pi, \gamma \bar{u} \rangle} e^{\epsilon \|W_\Pi\|} \frac{e^{-\langle W_\Pi, x_j y_j - \gamma \bar{u} \rangle}}{1 + e^{-\langle W_\Pi, (x_j + \delta_j) y_j \rangle}}$$

$$= \frac{\kappa}{n} \|\mathcal{P}_\perp W_\Pi\| e^{-\langle W_\Pi, \gamma \bar{u} \rangle} e^{\epsilon \|W_\Pi\|} \frac{e^{-\langle \mathcal{P}_\perp W_\Pi, \mathcal{P}_\perp x_j y_j \rangle}}{1 + e^{\epsilon \|W_\Pi\|} e^{-\langle \mathcal{P}_\perp W_\Pi, \mathcal{P}_\perp x_j y_j \rangle} e^{-\langle W_\Pi, \gamma \bar{u} \rangle}}$$

$$\geq \frac{\kappa}{2n} \|\mathcal{P}_\perp W_\Pi\| e^{-\langle W_\Pi, \gamma \bar{u} \rangle} e^{\epsilon \|W_\Pi\|}, \tag{59}$$

where we use that $\epsilon\|W_\Pi\|$, $\langle W_\Pi, \gamma\bar{u}\rangle$, $\langle \mathcal{P}_\perp W_\Pi, \mathcal{P}_\perp x_j y_j\rangle > 0$. Each term of the second part of (58) can be bounded by

$$
\frac{e^{-\langle W_\Pi, (x_k+\delta_k)y_k\rangle}}{1 + e^{-\langle W_\Pi, (x_k+\delta_k)y_k\rangle}} \langle -\mathcal{P}_\perp W_\Pi, \mathcal{P}_\perp x_k y_k\rangle
$$

$$
\geq e^{\epsilon\|W_\Pi\|} e^{-\langle W_\Pi, x_k y_k\rangle} \langle -\mathcal{P}_\perp W_\Pi, \mathcal{P}_\perp x_k y_k\rangle
$$

$$
= e^{\epsilon\|W_\Pi\|} e^{-\langle W_\Pi, \gamma\bar{u}\rangle} e^{-\langle W_\Pi, x_k y_k - \gamma\bar{u}\rangle} \langle -\mathcal{P}_\perp W_\Pi, \mathcal{P}_\perp x_k y_k\rangle
$$

$$
\geq e^{\epsilon\|W_\Pi\|} e^{-\langle W_\Pi, \gamma\bar{u}\rangle} e^{-\langle -\mathcal{P}_\perp W_\Pi, \mathcal{P}_\perp x_k y_k\rangle} \langle -\mathcal{P}_\perp W_\Pi, \mathcal{P}_\perp x_k y_k\rangle
$$

$$
\geq e^{\epsilon\|W_\Pi\|} e^{-\langle W_\Pi, \gamma\bar{u}\rangle} \left(-\frac{1}{e}\right), \tag{60}
$$

where the second inequality follows from

$$
\langle W_\Pi, x_k y_k - \gamma\bar{u}\rangle = \langle W_\Pi, \mathcal{P}_\perp x_k y_k\rangle + \langle W_\Pi, x_k y_k - \mathcal{P}_\perp x_k y_k - \gamma\bar{u}\rangle
$$

$$
\geq \langle W_\Pi, \mathcal{P}_\perp x_k y_k\rangle = \langle \mathcal{P}_\perp W_\Pi, \mathcal{P}_\perp x_k y_k\rangle
$$

while the last inequality comes from $f(x) = -xe^{-x} \geq -1/e$ for $x \geq 0$. Finally, we have

$$
\left\langle \mathcal{P}_\perp W_\Pi, (x_i + \delta_i)y_i \tilde{\ell}'(W_\Pi)\right\rangle \geq e^{-\langle W_\Pi, \gamma\bar{u}\rangle} e^{\epsilon\|W_\Pi\|} \left(\frac{\kappa\|\mathcal{P}_\perp W_\Pi\|}{2n} - \frac{1}{e}\right). \tag{61}
$$

Therefore, $\left\langle \mathcal{P}_\perp W_\Pi, (x_i + \delta_i)y_i \tilde{\ell}'(W_\Pi)\right\rangle \geq 0$ if

$$
\|\mathcal{P}_\perp W_\Pi\| \geq \frac{2n}{e\kappa} = \varphi. \tag{62}
$$

$\square$

We are now ready to prove the alignment phenomenon for the first layer. Let $\mathcal{P}_\perp W_1$ denote the projection of rows of $W_1$ onto $\bar{u}^\perp$. We start from exploring the asymptotic behavior of $\frac{\|\mathcal{P}_\perp W_1\|_F}{\|W_1\|_F}$ when $t \to \infty$. The update of $\mathcal{P}_\perp W_1(t+1)$ can be written explicitly as

$$
\|\mathcal{P}_\perp W_1(t+1)\|_F^2
$$

$$
= \|\mathcal{P}_\perp W_1(t)\|_F^2 - 2\eta^{(t)} \left\langle \mathcal{P}_\perp W_1(t), \mathcal{P}_\perp \frac{\partial\tilde{\mathcal{L}}}{\partial W_1(t)}\right\rangle + \eta^{(t)2} \left\|\mathcal{P}_\perp \frac{\partial\tilde{\mathcal{L}}}{\partial W_1(t)}\right\|_F^2
$$

$$
\leq \|\mathcal{P}_\perp W_1(t)\|_F^2 - 2\eta^{(t)} \left\langle \mathcal{P}_\perp W_1(t), \mathcal{P}_\perp \frac{\partial\tilde{\mathcal{L}}}{\partial W_1(t)}\right\rangle + \eta^{(t)2} \left\|\frac{\partial\tilde{\mathcal{L}}}{\partial W_1(t)}\right\|_F^2
$$

$$
\leq \|\mathcal{P}_\perp W_1(t)\|_F^2 - 2\eta^{(t)} \left\langle \mathcal{P}_\perp W_1(t), \frac{\partial\tilde{\mathcal{L}}}{\partial W_1(t)}\right\rangle + 2\left(\tilde{\mathcal{L}}(W(t+1)) - \tilde{\mathcal{L}}(W(t))\right) \tag{63}
$$

where the last inequality follows from Eq. (30). We define a large enough step $t_0$ as follows: for any $t \geq t_0$, we have

$$
\|W_1(t+1)\|_F^2 = \|W_1(t)\|_F^2 - 2\eta(t) \left\langle W_1(t), \frac{\partial\tilde{\mathcal{L}}}{\partial W_1(t)}\right\rangle + \eta(t)^2 \left\|\frac{\partial\tilde{\mathcal{L}}}{\partial W_1(t)}\right\|_F^2 \geq \|W_1(t)\|_F^2, \tag{64}
$$

where "large enough $t_0$" means that

$$
\frac{\varphi^2 + \tilde{\mathcal{L}}(t_0)}{\|W_1(t_0)\|_F} \to 0 \tag{65}
$$

as $\|W_1\|_F \to \infty$. Suppose that there exists a $t_1 \geq t_0$ such that

$$
\|\mathcal{P}_\perp W_\Pi(t_1 - 1)\|_F < \varphi \text{ and } \|\mathcal{P}_\perp W_\Pi(t_1)\|_F \geq \varphi, \tag{66}
$$

which is to say (recalling Lemma 8)

$$\left\langle \mathcal{P}_\perp W_1(t), \frac{\partial \tilde{\mathcal{L}}}{\partial W_1(t)} \right\rangle \geq 0$$

$$\implies \|\mathcal{P}_\perp W_1(t+1)\|_F^2 \leq \|\mathcal{P}_\perp W_1(t)\|_F^2 + 2\left(\tilde{\mathcal{L}}(W(t)) - \tilde{\mathcal{L}}(W(t+1))\right), \qquad (67)$$

for $t_0 \leq t_1 \leq t \leq t_2$ and $t_2 = \infty$ if we never have $\|\mathcal{P}_\perp W_\Pi(t)\|_F < \varphi$ after $t_1$. If there does not exist such a $t_1$, then we directly conclude that $\frac{\|\mathcal{P}_\perp W_1\|_F}{\|W_1\|_F} \to 0$ since $\|W_1\|_F \to \infty$. On the other hand, for any $t_0 \leq t_1 \leq t \leq t_2$

$$\begin{aligned}
\frac{\|\mathcal{P}_\perp W_1(t)\|_F^2}{\|W_1(t)\|_F^2} &\leq \frac{\|\mathcal{P}_\perp W_1(t_1)\|_F^2 + 2\left(\tilde{\mathcal{L}}(W(t_1)) - \tilde{\mathcal{L}}(W(t))\right)}{\|W_1(t_1)\|_F^2} \\
&\leq \frac{\|\mathcal{P}_\perp W_1(t_1)\|_F^2 + 2\tilde{\mathcal{L}}(W(t_1))}{\|W_1(t_1)\|_F^2} \\
&\leq \frac{\varphi^2 + 2\mu(t_1)\varphi + \mu^2(t_1) + 2\tilde{\mathcal{L}}(W(t_1))}{\|W_1(t_1)\|_F^2} \\
&= \frac{\Phi(\varphi, t_1)}{\|W_1(t_1)\|_F^2}
\end{aligned} \qquad (68)$$

where the last inequality follows from Eq. (31) and $\Phi(\varphi, t_1)$ is defined by

$$\Phi(\varphi, t_1) = \varphi^2 + 2\mu(t_1)\varphi + \mu^2(t_1) + 2\tilde{\mathcal{L}}(W(t_1)) \geq \Phi(\varphi, t') \qquad (69)$$

for any $t' \geq t_1$ because the loss and $\mu$ never increase. Therefore we can conclude that

$$\frac{\|\mathcal{P}_\perp W_1(t)\|_F^2}{\|W_1(t)\|_F^2} \leq \frac{\Phi(\varphi, t)}{\|W_1(t)\|_F^2} \leq \frac{\Phi(\varphi, t_1)}{\|W_1(t_1)\|_F^2} \to 0 \qquad (70)$$

for any $t \geq t_1$. As a result,

$$|\langle v_1(t), \bar{u}\rangle| \to 1. \qquad (71)$$

## B    PROOFS OF SECTION 3.2

We present some useful properties and proofs of Lemma 3 and Lemma 4 in Section B.1. Section B.2 focuses on the divergences of weight norms. Section B.3 is about the convergence to KKT points(Theorem 5).

**Additional notations**    We use $W_{ij;k}$ to represent the $i$-th row $j$-th column entry of the $k$-th layer weight martrix.

### B.1    USEFUL PROPERTIES AND PROOFS OF LEMMA 3 AND LEMMA 4

For any $m \times n$ matrix $A$, we denote $v(A)$, an $mn$-dimensional vector, as the vectorized version of it:

$$v(A) = \begin{pmatrix} A_{11} \\ A_{21} \\ \cdots \\ A_{m1} \\ \cdots \\ A_{mn} \end{pmatrix}, \qquad (72)$$

then the trace operator can be represented by

$$\mathrm{tr}\left(A^T B\right) = v(A)^T v(B) = v(B)^T v(A). \qquad (73)$$

We first introduce the Euler's theorem on homogeneous functions.

**Lemma 9** (Euler's theorem on homogeneous functions). *If $f(W_1, \ldots, W_L)$ is a positive multi-$c_k$-homogeneous function*

$$f(\rho_1 W_1, \ldots, \rho_L W_L) = \prod_{k=1}^{L} \rho_k^{c_k} f(W_1, \ldots, W_L) \tag{74}$$

*for positive constants $\rho_k$'s and $c_k \geq 1$, then we have:*

$$tr\left(\frac{\partial f(W_1, \ldots, W_L)}{\partial W_k} W_k\right) = c_k f(W_1, \ldots, W_L). \tag{75}$$

*Proof.* Taking derivatives of $\rho_k$ on both sides of Eq. (74) for any given $k \in \{1, \ldots, L\}$, we have

$$\left\langle W_k, \frac{\partial f}{\partial W_k} \right\rangle = tr\left(W_k \frac{\partial f}{\partial W_k}\right) = \prod_{k' \neq k}^{L} \rho_{k'}^{c_{k'}} f(x; W) c_k \rho_k^{c_k - 1}. \tag{76}$$

Since $\rho_k$ is arbitrary, we let $\rho_L = \cdots = \rho_k = \cdots = \rho_1 = 1$, then the above equation becomes

$$tr\left(W_k \frac{\partial f(W_1, \ldots, W_L)}{\partial W_k}\right) = c_k f(W_1, \ldots, W_L) \tag{77}$$

for any $k \in \{1, \ldots, L\}$. $\qquad\square$

Furthermore, we have

$$\left\langle \frac{\partial f(x; W)}{\partial W}, W \right\rangle = \sum_{k=1}^{L} tr\left(\frac{\partial f(x; W)}{\partial W_k} W_k\right)$$
$$= K f(x, W) \tag{78}$$

where $K = \sum_{k=1}^{L} c_k$. For the deep neural networks defined in Eq. (2) with homogeneous property, we can then apply the above lemma to them.

We examine whether Assumption 2 can still be applied to adversarial margin under Assumption 1.

**Lemma 10** (Assumption 2 for adversarial training). *For any fixed $x$,*

- *if $yf(x; W)$ is locally Lipschitz, then so does $yf(x + \delta(W); W)$ with perturbation $\delta(W)$.*

  *Proof.* For fixed $x$, suppose $yf(W)$ is locally Lipschitz on $Y$, then for each $W \subset Y$ there is a $Z_W$ containing $W$ such that $yf(W)$ is Lipschitz on $Z_W$:

  $$\|yf(W) - yf(V)\| \leq L_W \|W - V\| \text{ for } W, V \subset Z_W. \tag{79}$$

For the adversarial perturbation, by definition $\delta(W)$ and $\delta(V)$ are solutions of the inner maximization for the adversarial training, in other words,

$$yf(x + \delta(W); W) \leq yf(x + \delta(V); W) \tag{80}$$
$$yf(x + \delta(V); V) \leq yf(x + \delta(W); V) \tag{81}$$

because then we will have $\ell(x + \delta(W); W) > \ell(x + \delta(V); W)$ since $\ell$ is non-increasing under Assumption 1. As a result,

$$yf(x + \delta(W); W) - yf(x + \delta(V); V) \leq yf(x + \delta(V); W) - yf(x + \delta(V); V)$$
$$\leq L_W \|W - V\| \tag{82}$$

for $yf(x + \delta(W); W) > yf(x + \delta(V); V)$. The result for $yf(x + \delta(W); W) < yf(x + \delta(V); V)$ is similar. $\qquad\square$

- *if $\forall i : y_i f(x_i; W)$ have locally Lipschitz gradients, then so do $y_i f(x_i + \delta_i(W); W)$.*

*Proof.* For fixed $x$, suppose $y\nabla f(W)$ is locally Lipschitz on $Y$, then for each $W \subset Y$ there is a $Z_W$ containing $W$ such that $y\nabla f(W), \delta(W)$ are Lipschitz on $Z_W$, for $W, V \subset Z_W$:

$$\|y\nabla f(x; W) - y\nabla f(x; V)\| \le L_W \|W - V\| \tag{83}$$

$$\|y\nabla f(x + \delta(W); W) - y\nabla f(x + \delta(V); W)\| \le L_{Wx}\|\delta(W) - \delta(V)\|$$
$$\le L_{Wx}L_{W\delta}\|W - V\| \tag{84}$$

Then

$$\|y_i\nabla f(x_i + \delta_i(W); W) - y_i\nabla f(x_i + \delta_i(V); V)\|$$
$$\le \|y_i\nabla f(x_i + \delta_i(W); W) - y_i\nabla f(x_i + \delta_i(W); V)\|$$
$$+ \|y_i\nabla f(x_i + \delta_i(W); V) - y_i\nabla f(x_i + \delta_i(V); V)\|$$
$$\le (L_W + L_{Wx}L_{W\delta}) \|W - V\|. \tag{85}$$

$\square$

### B.1.1 PROOF OF LEMMA 3

*Proof.* The proof can be done for multi-$c_k$-homogeneous networks as defined in Lemma 9. We first note that

$$\nabla_x f(x; \rho_1 W_1, \ldots, \rho_L W_L) = \prod_{k=1}^{L} \rho_k^{c_k} \nabla_x f(x; W_1, \ldots, W_L) \tag{86}$$

by taking derivatives with respect to $x$ on both sides of Eq. (74). Therefore $\nabla_x f(x; W_1, \ldots, W_L)$ is also positive homogeneous. Under Assumption 1, we prove Lemma 3 in the following cases:

1. $\ell_2$-FGM perturbation. This perturbations is taken as

$$\delta_{\text{FGM}}(W) = \frac{\epsilon y\tilde{\ell}'\nabla_x f(x; W_1, \ldots, W_L)}{\left\|y\tilde{\ell}'\nabla_x f(x; W_1, \ldots, W_L)\right\|} = -\frac{\epsilon y\nabla_x f(x; W_1, \ldots, W_L)}{\|\nabla_x f(x; W_1, \ldots, W_L)\|} \tag{87}$$

because $\tilde{\ell}' < 0$. Invoking Eq. (86), we know that

$$\delta_{\text{FGM}}(\rho_1 W_1, \ldots, \rho_L W_L) = -\frac{\epsilon y \prod_{k=1}^{L} \rho_k^{c_k} \nabla_x f(x; W_1, \ldots, W_L)}{\prod_{k=1}^{L} \rho_k^{c_k} \|\nabla_x f(x; W_1, \ldots, W_L)\|}$$
$$= -\frac{\epsilon y \nabla_x f(x; W_1, \ldots, W_L)}{\|\nabla_x f(x; W_1, \ldots, W_L)\|}$$
$$= \delta_{\text{FGM}}(W_1, \ldots, W_L). \tag{88}$$

2. FGSM perturbation.

$$\delta_{\text{FGSM}}(W) = \epsilon \text{sgn}\left(y\tilde{\ell}'\nabla_x f(x; W_1, \ldots, W_L)\right), \tag{89}$$

where sgn() denotes the sign function. Then we have

$$\delta_{\text{FGSM}}(\rho_1 W_1, \ldots, \rho_L W_L) = \epsilon \text{sgn}\left(y\tilde{\ell}' \prod_{k=1}^{L} \rho_k^{c_k} \nabla_x f(x; W_1, \ldots, W_L)\right) = \delta_{\text{FGSM}}(W) \tag{90}$$

since $\rho_k > 0$ for any $k$ thus will not affect the sign operation.

3. $\ell_2$ and $\ell_\infty$-PGD perturbation. We only prove the case for $\ell_2$-PGD and the case for $\ell_\infty$-PGD is similar.

$$\delta_{\text{PGD}}^{j+1}(W) = \mathcal{P}_{\mathcal{B}(0,\epsilon)}\left[\delta_{\text{PGD}}^{j}(W) - \xi\frac{y\nabla_x f(x; W_1, \ldots, W_L)}{\|\nabla_x f(x; W_1, \ldots, W_L)\|}\right], \tag{91}$$

where $\mathcal{P}_{\mathcal{B}(0,\epsilon)}$ denotes the projection operator, $\mathcal{B}(0,\epsilon)$ is the perturbation set, $j$ is the step indices of PGD and $\xi$ is the step size. We prove by induction. For $j = 0$, we have

$$\delta^1_{\text{PGD}}(\rho_1 W_1, \ldots, \rho_L W_L) = \mathcal{P}_{\mathcal{B}(0,\epsilon)} \left[ -\xi \frac{y\nabla_x f(x; W_1, \ldots, W_L)}{\|\nabla_x f(x; W_1, \ldots, W_L)\|} \right]$$
$$= \delta^1_{\text{PGD}}(W_1, \ldots, W_L). \tag{92}$$

If we have $\delta^j_{\text{PGD}}(\rho_1 W_1, \ldots, \rho_L W_L) = \delta^j_{\text{PGD}}(W_1, \ldots, W_L)$, then for $j + 1$, we have

$$\delta^{j+1}_{\text{PGD}}(\rho_1 W_1, \ldots, \rho_L W_L) = \mathcal{P}_{\mathcal{B}(0,\epsilon)} \left[ \delta^j_{\text{PGD}}(\rho W) - \xi \frac{y \prod_{k=1}^L \rho_k \nabla_x f(x; W_1, \ldots, W_L)}{\prod_{k=1}^L \rho_k \|\nabla_x f(x; W_1, \ldots, W_L)\|} \right]$$
$$= \mathcal{P}_{\mathcal{B}(0,\epsilon)} \left[ \delta^j_{\text{PGD}}(W) - \xi \frac{y\nabla_x f(x; W_1, \ldots, W_L)}{\|\nabla_x f(x; W_1, \ldots, W_L)\|} \right]$$
$$= \delta^{j+1}_{\text{PGD}}(W_1, \ldots, W_L). \tag{93}$$

Therefore, these four adversarial perturbations are all scale invariant adversarial perturbations by our definition. $\qquad\square$

### B.1.2 PROOF OF LEMMA 4

*Proof.* This can be easily proved by noting that, for any multi-$c_k$-homogeneous functions,

$$f(x_i + \delta_i(\rho_1 W_1, \ldots, \rho_L W_L); \rho_1 W_1, \ldots, \rho_L W_L) = \prod_{k=1}^L \rho_k^{c_k} f(x_i + \delta_i(W_1, \ldots, W_L); W_1, \ldots, W_L)$$
$$\tag{94}$$

because $\delta(W)$ is a scale invariant adversarial perturbations. Then $f(x + \delta(W); W)$ is still a homogeneous function and Lemma 9 can be applied to $f(x + \delta(W); W_1, \ldots, W_L)$ and we have

$$\text{tr}\left( \frac{\partial \tilde{\mathcal{L}}}{\partial W_k} W_k \right) = \frac{1}{n} \sum_{i=1}^n \tilde{\ell}'_i y_i \text{tr}\left( \frac{\partial f(x_i + \delta_i(W); W_1, \ldots, W_L)}{\partial W_k} W_k \right)$$
$$= -\frac{c_k}{n} \sum_{i=1}^n e^{-\tilde{\gamma}_i} y_i f(x_i + \delta_i(W); W_1, \ldots, W_L)$$
$$= -\frac{c_k}{n} \sum_{i=1}^n e^{-\tilde{\gamma}_i} \tilde{\gamma}_i. \tag{95}$$

Taking $c_1 = \cdots = c_L$ gives us Lemma 4. $\qquad\square$

### B.2 DIVERGENCES OF FROBENIUS NORMS OF WEIGHT MATRICES

### B.2.1 DIVERGENCES OF ALL LAYERS

**Lemma 11.** *The Frobenius norms of all layers grow at approximately the same rate along the trajectory of gradient flow for adversarial training with scale invariant adversarial perturbations for multi-$c_k$-homogeneous DNNs*

$$\frac{1}{c_L} \frac{d\|W_L\|_F^2}{dt} = \frac{1}{c_{L-1}} \frac{d\|W_{L-1}\|_F^2}{dt} = \cdots = \frac{1}{c_1} \frac{d\|W_1\|_F^2}{dt} \tag{96}$$

**Remark.** Note that a similar conclusion as that of Lemma 11 exists for gradient flow of the standard training for multi-1-homogeneous networks Du et al. (2018). We generalize this property to the adversarial training of multi-$c_k$-homogeneous neural networks. When $c_1 = \cdots = c_L$, we have that all layer grow at the same rate.

*Proof.* For any $W_k$,

$$
\frac{d\|W_k\|_F^2}{dt} = \sum_{i,j} \frac{d\left(W_{ij;k} W_{ji;k}^T\right)}{dt} = -2 \sum_{i,j} \left(\frac{\partial \tilde{\mathcal{L}}}{\partial W_{ij;k}}\right)^T W_{ji;k}^T
$$
$$
= \frac{2c_k}{n} \sum_{i=1}^{n} e^{-\tilde{\gamma}_i} \dot{\tilde{\gamma}}_i, \tag{97}
$$

where we use $\frac{dW}{dt} = -\left(\frac{\partial \tilde{\mathcal{L}}}{\partial W_k}\right)^T$ and Lemma 4. One can then immediately notice that the above equation does not depend on any specific $k$, thus we have

$$
\frac{1}{c_L} \frac{d\|W_L\|_F^2}{dt} = \frac{1}{c_{L-1}} \frac{d\|W_{L-1}\|_F^2}{dt} = \cdots = \frac{1}{c_1} \frac{d\|W_1\|_F^2}{dt}. \tag{98}
$$

$\square$

### B.2.2 PROOF OF THEOREM 3

We prove this theorem by exploring the time evolution of $\rho$.

*Proof.* We start from a multi-$c_k$-homogeneous function then take $c_1 = \cdots = c_L$. Recalling $f(x; W) = \rho f(x; \widehat{W})$ where $\widehat{W}_k = W_k / \|W_k\|_F$ and

$$
\rho = \prod_{k=1}^{L} \rho_k^{c_k} = \prod_{k=1}^{L} \|W_k\|_F^{c_k}, \tag{99}
$$

the adversarially trained predictor with scale invariant adversarial perturbation is still homogeneous (see Eq. (94)):

$$
f\left(x + \delta(W); W\right) = \rho f\left(x + \delta(\widehat{W}); \widehat{W}\right). \tag{100}
$$

For $\rho_k = \|W_k\|_F$, we have

$$
\frac{d\rho_k}{dt} = \frac{c_k}{n\rho_k} \sum_{i=1}^{n} e^{-\tilde{\gamma}_i} \dot{\tilde{\gamma}}_i \tag{101}
$$

by invoking Eq. (97). The dynamical evolution of $\rho$ will then be

$$
\frac{d\rho}{dt} = \sum_{k=1}^{L} \rho_1^{c_1} \cdots \frac{d\rho_k^{c_k}}{dt} \cdots \rho_L^{c_L} = \sum_{k=1}^{L} \frac{c_k^2 \rho^2}{\rho_k^2} \left(\frac{1}{n} \sum_i e^{-\tilde{\gamma}_i} \dot{\tilde{\gamma}}_i\right). \tag{102}
$$

Let $t_0$ denote the time such that all worst case adversarial examples are correctly classified. We study the trends for $\rho$ after $t_0$. On one hand, the empirical adversarial loss does not increase:

$$
-\frac{d\tilde{\mathcal{L}}}{dt} = -\frac{\partial \tilde{\mathcal{L}}}{\partial W} \frac{dW}{dt} = \left\|\frac{dW}{dt}\right\|_2^2 \geq 0. \tag{103}
$$

On the other hand, recalling our definition of normalized adversarial margin, let

$$
m = \arg \min_{i \in \{1,\ldots,n\}} \tilde{\gamma}_i = \arg \min_{i \in \{1,\ldots,n\}} \rho \hat{\gamma}_i \tag{104}
$$

and note the definition of $t_0$, we have

$$
e^{-\tilde{\gamma}_m} \leq \frac{1}{n} \sum_i e^{-\tilde{\gamma}_i} = \tilde{\mathcal{L}} \implies \tilde{\gamma}_m \geq \ln\left(\frac{1}{\tilde{\mathcal{L}}(t)}\right) \geq \ln\left(\frac{1}{\tilde{\mathcal{L}}(t_0)}\right) > 0 \tag{105}
$$

because $\tilde{\mathcal{L}}$ does not increase and $\tilde{\mathcal{L}}(t) \leq \tilde{\mathcal{L}}(t_0) < 1$. This also implies that $\tilde{\gamma}_m$, thus $\tilde{\gamma}_i$ for all $i \in \{1, \ldots, n\}$, can not be arbitrarily close to 0. Otherwise one would conclude that $\tilde{\mathcal{L}}(t)$ may be arbitrarily close to 1 which is a contradiction. Therefore we can immediately conclude that $d\rho/dt > 0$ in Eq. (102), which implies that $\rho$ may diverge as $t \to \infty$. To have a clearer view

about the convergence property of $\rho$, we study the following relation regarding the time evolution of $e^{\rho y_i f(x_i + \delta_i(\widehat{W}))}$ after $t_0$:

$$\sum_i \frac{de^{\rho y_i f(x_i + \delta_i(\widehat{W}); \widehat{W})}}{dt} = \underbrace{\sum_i e^{\rho \hat{\gamma}_i} \hat{\gamma}_i \frac{d\rho}{dt}}_{\text{(a)}} + \underbrace{\sum_i e^{\rho \hat{\gamma}_i} \rho y_i \frac{df(x_i + \delta_i(\widehat{W}); \widehat{W})}{dt}}_{\text{(b)}}. \qquad (106)$$

The term (a) can be computed by invoking Eq. (102)

$$\begin{aligned}
\text{(a)} &= \sum_{k=1}^{L} \frac{c_k^2 \rho^2}{n \rho_k^2} \sum_i e^{\rho \hat{\gamma}_i} \hat{\gamma}_i \left( \sum_j e^{-\rho \hat{\gamma}_j} \hat{\gamma}_j \right) \\
&= \sum_{k=1}^{L} \frac{c_k^2 \rho^2}{n \rho_k^2} \left( \sum_{i=1}^{n} \hat{\gamma}_i^2 + \frac{1}{2} \sum_{i \neq j}^{n} \hat{\gamma}_i \hat{\gamma}_j \left( e^{\rho(\hat{\gamma}_i - \hat{\gamma}_j)} + e^{-\rho(\hat{\gamma}_i - \hat{\gamma}_j)} \right) \right) \\
&\geq \sum_{k=1}^{L} \frac{c_k^2 \rho^2}{n \rho_k^2} \left( \sum_{i=1}^{n} \hat{\gamma}_i \right)^2 \geq \sum_{k=1}^{L} \frac{\rho^2 \zeta^2}{n \rho_k^2},
\end{aligned} \qquad (107)$$

where the first inequality follows from $x + 1/x \geq 2$ for $x \geq 0$ and we denote the minimum of $\sum_i \hat{\gamma}_i$ for $t \geq t_0$ as $\zeta > 0$. The existence of such $\zeta$ has been discussed below Eq. (105).

The term (b) needs more analysis. For any fixed adversarial example $x_i + \delta(\widehat{W})$, we have

$$y_i \frac{df(x_i + \delta_i(\widehat{W}); \widehat{W})}{dt} = y_i \sum_{k=1}^{L} \text{tr} \left( \frac{\partial f(x_i + \delta_i(\widehat{W}); \widehat{W})}{\partial \widehat{W}_k} \frac{d\widehat{W}_k}{dt} \right), \qquad (108)$$

where $d\widehat{W}_k/dt$ can be computed as follows

$$\begin{aligned}
\frac{d\widehat{W}_{jl;k}}{dt} &= \sum_{m,n} \frac{\partial \widehat{W}_{jl;k}}{\partial W_{mn;k}} \frac{dW_{mn;k}}{dt} \\
&= \sum_{m,n} \frac{\partial}{\partial W_{mn;k}} \left( \frac{W_{jl;k}}{\sqrt{\text{tr}(W_k W_k^T)}} \right) \frac{dW_{mn;k}}{dt} \\
&= \frac{1}{\rho_k} \left( \frac{dW_{jl;k}}{dt} - \frac{\widehat{W}_{jl;k}}{n\rho_k} \sum_{i=1}^{n} y_i e^{-\tilde{\gamma}_i} \tilde{\gamma}_i \right).
\end{aligned} \qquad (109)$$

On the other hand, we note that

$$\begin{aligned}
\frac{dW_k}{dt} &= -\sum_i \frac{\partial \tilde{\mathcal{L}}}{\partial f(x_i + \delta_i(W); W)} \left( \frac{\partial f(x_i + \delta_i(W); W)}{\partial W_k} \right)^T \\
&= \frac{\rho}{n\rho_k} \sum_j y_j e^{-\tilde{\gamma}_j} \left( \frac{\partial f(x_j + \delta_j(\widehat{W}); \widehat{W})}{\partial \widehat{W}_k} \right)^T
\end{aligned} \qquad (110)$$

because $\frac{\partial f(W)}{\partial W_k} = \frac{\rho}{\rho_k} \frac{\partial f(\widehat{W})}{\partial \widehat{W}_k}$. Combing Eq. (109) and Eq. (110), we have

$$\frac{d\widehat{W}_k}{dt} = \frac{\rho}{n\rho_k^2} \sum_j y_j e^{-\tilde{\gamma}_j} \left[ \left( \frac{\partial f(x_j + \delta_j(\widehat{W}); \widehat{W})}{\partial \widehat{W}_k} \right)^T - \widehat{W}_k f(x_j + \delta_j(\widehat{W}); \widehat{W}) \right]. \qquad (111)$$

Therefore

$$v \left( \left( \frac{d\widehat{W}_k}{dt} \right)^T \right) = \frac{\rho}{n\rho_k^2} \sum_j y_j e^{-\tilde{\gamma}_j} A_k v \left( \frac{\partial f(x_j + \delta_j(\widehat{W}); \widehat{W})}{\partial \widehat{W}_k} \right), \qquad (112)$$

where

$$A_k = I - \frac{v\left(W_k^T\right) v\left(W_k^T\right)^T}{\|W_k\|_F^2} = I - v\left(\widehat{W}_k^T\right) v\left(\widehat{W}_k^T\right)^T \tag{113}$$

can be seen as a projector operator. Putting Eq. (112) into Eq. (108) will give us the expression

$$y_i \frac{df\left(x_i + \delta_i(\widehat{W}); \widehat{W}\right)}{dt} = \sum_{k=1}^{L} \frac{\rho}{n\rho_k^2} \sum_j e^{-\tilde{\gamma}_j} v\left(\frac{\partial \hat{\gamma}_j(\widehat{W})}{\partial \widehat{W}_k}\right)^T A_k^T v\left(\frac{\partial \hat{\gamma}_i(\widehat{W})}{\partial \widehat{W}_k}\right). \tag{114}$$

Now the term (b) in Eq. (106) is, by putting Eq. (114) back into it,

$$(\text{b}) = \sum_{k=1}^{L} \frac{\rho^2}{n\rho_k^2} \sum_i e^{\rho\hat{\gamma}_i} \left[\sum_j e^{-\rho\hat{\gamma}_j} v\left(\frac{\partial \hat{\gamma}_j}{\partial \widehat{W}_k}\right)^T A_k^T v\left(\frac{\partial \hat{\gamma}_i}{\partial \widehat{W}_k}\right)\right]$$

$$= \frac{1}{2} \sum_{k=1}^{L} \frac{\rho^2}{n\rho_k^2} \underbrace{\sum_{i,j=1}^{n} v\left(\frac{\partial \hat{\gamma}_i}{\partial \widehat{W}_k}\right)^T A_k^T A_k v\left(\frac{\partial \hat{\gamma}_j}{\partial \widehat{W}_k}\right) \left[e^{\rho(\hat{\gamma}_i - \hat{\gamma}_j)} + e^{-\rho(\hat{\gamma}_i - \hat{\gamma}_j)}\right]}_{(\text{c})}. \tag{115}$$

Rearranging $e^{\rho(\hat{\gamma}_i - \hat{\gamma}_j)} + e^{-\rho(\hat{\gamma}_i - \hat{\gamma}_j)}$ as follows

$$e^{\rho(\hat{\gamma}_i - \hat{\gamma}_j)} + e^{-\rho(\hat{\gamma}_i - \hat{\gamma}_j)} = e^{\rho(\hat{\gamma}_i + \hat{\gamma}_j)} + e^{-\rho(\hat{\gamma}_i + \hat{\gamma}_j)} - (e^{\rho\hat{\gamma}_j} - e^{-\rho\hat{\gamma}_j})(e^{\rho\hat{\gamma}_i} - e^{-\rho\hat{\gamma}_i})$$

will allow us to rewrite (c) as

$$(\text{c}) = \left\|\sum_i e^{\rho\hat{\gamma}_i} A_k v\left(\frac{\partial \hat{\gamma}_i}{\partial \widehat{W}_k}\right)\right\|_2^2 + \left\|\sum_i e^{-\rho\hat{\gamma}_i} A_k v\left(\frac{\partial \hat{\gamma}_i}{\partial \widehat{W}_k}\right)\right\|_2^2$$

$$- \left\|\sum_i \left(e^{\rho\hat{\gamma}_i} - e^{-\rho\hat{\gamma}_i}\right) A_k v\left(\frac{\partial \hat{\gamma}_i}{\partial \widehat{W}_k}\right)\right\|_2^2$$

$$= \left\|\sum_i e^{\rho\hat{\gamma}_i} A_k v\left(\frac{\partial \hat{\gamma}_i}{\partial \widehat{W}_k}\right)\right\|_2^2 + \left\|\sum_i e^{-\rho\hat{\gamma}_i} A_k v\left(\frac{\partial \hat{\gamma}_i}{\partial \widehat{W}_k}\right)\right\|_2^2$$

$$- \left\|\sum_i e^{\rho\hat{\gamma}_i} A_k v\left(\frac{\partial \hat{\gamma}_i}{\partial \widehat{W}_k}\right) - \sum_i e^{-\rho\hat{\gamma}_i} A_k v\left(\frac{\partial \hat{\gamma}_i}{\partial \widehat{W}_k}\right)\right\|_2^2$$

$$\geq 0. \tag{116}$$

Combing Eq. (107) and Eq. (116) with Eq. (106), we will have

$$\frac{1}{n} \sum_i \frac{de^{\rho y_i f(x_i + \delta_i(\widehat{W}); \widehat{W})}}{dt} \geq \sum_{k=1}^{L} \frac{\rho^2(t)\zeta^2}{n^2 \rho_k^2(t)}. \tag{117}$$

According to Eq. (101), we know that $\forall k: d\rho_k/dt > 0$ after $t_0$, therefore $\forall k: \rho^2/\rho_k^2$ is lower bounded by its value at $t_0$ for $t \in [t_0, \infty)$ because

$$\frac{d}{dt}\left(\frac{\rho^2}{\rho_k^2}\right) = \frac{d}{dt}\left(\rho_k^{2c_k - 2} \prod_{k' \neq k}^{L} \rho_{k'}^{2c_{k'}}\right)$$

$$= 2(c_k - 1)\rho_k^{2c_k - 3} \dot\rho_k \prod_{k' \neq k}^{L} \rho_{k'}^{2c_{k'}} + \rho_k^{2c_k - 2} \sum_{k' \neq k}^{L} 2c_{k'} \rho_{k'}^{2c_{k'} - 1} \dot\rho_{k'} \left(\prod_{k'' \neq k', k}^{L} \rho_{k''}^{2c_{k''}}\right) > 0, \tag{118}$$

where $\dot\rho_k$ denotes $d\rho_k/dt > 0$ for any $k$ due to Eq. (101). Integrating both sides of Eq. (117) from $t_0$ to $t$ gives us

$$\frac{1}{n} \sum_i \left(e^{\rho(t)y_i f(x_i + \delta_i(\widehat{W}); \widehat{W}(t))} - e^{\rho(t_0)y_i f(x_i + \delta_i(\widehat{W}); \widehat{W}(t_0))}\right) \geq \sum_{k=1}^{L} \int_{t_0}^{t} \frac{\rho^2(\tau)\zeta^2}{n^2 \rho_k^2(\tau)} d\tau$$

$$\geq \sum_{k=1}^{L} \frac{\rho^2(t_0)\zeta^2}{n^2 \rho_k^2(t_0)}(t - t_0), \tag{119}$$

where the last inequality follows from Eq. (118). Let $t \to \infty$ in the above equation, we will have

$$\rho = \Omega\left(\ln t\right) \to \infty \tag{120}$$

since $y_i f(x_i + \delta_i(\widehat{W}); \widehat{W})$ is upper bounded, which can be easily deduced considering that all weights have norm 1, and $\zeta$ is lower bounded as discussed before. Taking $c_1 = \cdots = c_L$ gives us the conclusion. □

Note that $\rho = \rho_1^{c_1} \cdots \rho_L^{c_L}$ and $\rho_k$ for any $k$ grows at the same rate for multi-$c$-homogeneous functions (Lemma 11), we immediately have

**Corollary 1.** $\forall k \in \{1, \ldots, L\} : \rho_k \to \infty$ *as* $t \to \infty$.

### B.3 CONVERGENCE TO KKT POINT

#### B.3.1 PROOF OF THEOREM 4

*Proof.* Most calculations needed for the proof of this theorem have been done in Section B.2.2. According to Eq. (114), we have

$$\sum_i e^{-\tilde{\gamma}_i} y_i \frac{df(x + \delta_i(\widehat{W}); \widehat{W})}{dt} \propto \sum_{k=1}^{L} \frac{1}{\rho_k^2} \sum_i e^{-\tilde{\gamma}_i} \left( \sum_j e^{-\tilde{\gamma}_j} v \left( \frac{\partial \hat{\gamma}_j(\widehat{W})}{\partial \widehat{W}_k} \right)^T A_k^T v \left( \frac{\partial \hat{\gamma}_i(\widehat{W})}{\partial \widehat{W}_k} \right) \right)$$

$$= \sum_{k=1}^{L} \frac{1}{\rho_k^2} \left\| \sum_i e^{-\tilde{\gamma}_i} A_k v \left( \frac{\partial \hat{\gamma}_i}{\partial \widehat{W}_k} \right) \right\|_2^2 \geq 0. \tag{121}$$

□

#### B.3.2 PROOF OF THEOREM 5

The KKT conditions for the optimization problem Eq. (19) are

$$\forall k : \quad \frac{1}{2} \frac{\partial \rho_k^2}{\partial W_k} - \sum_{i=1}^{n} \lambda_i \frac{\partial \tilde{\gamma}_i}{\partial W_k} = 0 \tag{122}$$

$$\forall i \in \{1, \ldots, n\} : \lambda_i(\tilde{\gamma}_i - 1) = 0. \tag{123}$$

Following Dutta et al. (2013); Lyu & Li (2020), we define the *approximate KKT point* in our settings

**Definition 3** (Adapted from Definition C.4 in Lyu & Li (2020)). *The approximate KKT points of the optimization problem (19) are those feasible points which satisfy the following condtions:*

1. $\left\| \frac{1}{2} \frac{\partial \|W\|^2}{\partial W} - \sum_{i=1}^{n} \lambda_i \frac{\partial \tilde{\gamma}_i}{\partial W} \right\|_2 \leq \chi$;

2. $\forall i \in \{1, \ldots, n\} : \lambda_i(\tilde{\gamma}_i - 1) \leq \xi$,

*where $\chi, \xi > 0$ and $\lambda_i \geq 0$. These points are said to be $(\chi, \xi)$-approximate KKT points.*

We now present the proof of Theorem 5.

*Proof.* Let

$$\|W\| = \sqrt{\sum_{k=1}^{L} \rho_k^2}, \tag{124}$$

$$V = \frac{W}{\tilde{\gamma}_m^{1/K}}, \tag{125}$$

where $K = \sum_{k=1}^{L} c = cL$. Since

$$cf(x; W) = c\tilde{\gamma}_m f(x; V) \tag{126}$$

according to the definition of the multi-$c$-homogeneous functions and

$$cf(x; W) = \left\langle W_k, \frac{\partial f(x; W)}{\partial W_k} \right\rangle, cf(x; V) = \left\langle V_k, \frac{\partial f(x; V)}{\partial V_k} \right\rangle, \quad (127)$$

we have

$$\frac{\partial f(x; V)}{\partial V} = \frac{1}{\tilde{\gamma}_m^{(K-1)/K}} \frac{\partial f(x; W)}{\partial W}. \quad (128)$$

Specifically, in the following, we will show that $V$ is a $(\chi, \xi)$-approximate KKT points of the optimization problem (19) with $\chi, \xi \to 0$ along the training trajectory. Because the homogeneous property of the network trained with scale invariant adversarial perturbations, it satisfies the MFCQ condition by simply noting that

$$\forall i \in \{1, \ldots, n\} : \operatorname{tr}\left( \frac{\partial \tilde{\gamma}_i}{\partial W_k} W_k \right) = c\tilde{\gamma}_i > 0. \quad (129)$$

This makes a point being a KKT point a necessary condition for it to be the optimal solution.

We now show that the limit point of $V$ along the adversarial training trajectory is an $(\chi, \xi)$-approximate KKT point with $\chi, \xi \to 0$ starting with the first condition. Let

$$\lambda_i = \left( \left\| \frac{\partial \tilde{\mathcal{L}}}{\partial W} \right\| \right)^{-1} \|W\| \tilde{\gamma}_m^{(K-2)/K} e^{-\tilde{\gamma}_i}, \quad (130)$$

then we have

$$\left\| V - \sum_i \lambda_i y_i \frac{\partial f(x_i + \delta_i(V); V)}{\partial V} \right\|_2^2 \overset{\mathsf{a}}{=} \left\| \frac{W}{\tilde{\gamma}_m^{1/K}} - \sum_i \frac{\lambda_i y_i}{\tilde{\gamma}_m^{(K-1)/K}} \frac{\partial f(x_i + \delta_i(W); W)}{\partial W} \right\|_2^2$$

$$= \frac{2(\sum_k \rho_k^2)}{(\prod_k \rho_k^{2c})^{1/K} \hat{\gamma}_m^{2/K}} \left( 1 - \frac{\left\langle W, \frac{\partial \tilde{\mathcal{L}}}{\partial W} \right\rangle}{\|W\| \left\| \frac{\partial \tilde{\mathcal{L}}}{\partial W} \right\|^2} \right)$$

$$\leq \frac{2L\rho_{k'}^2}{\rho_{k''}^2 \hat{\gamma}_m^{2/K}} \left( 1 - \frac{\left\langle W, \frac{\partial \tilde{\mathcal{L}}}{\partial W} \right\rangle}{\|W\| \left\| \frac{\partial \tilde{\mathcal{L}}}{\partial W} \right\|^2} \right) = \chi(t) \quad (131)$$

where $\mathsf{a}$ follows from (128); $k' = \arg\max_k \rho_k$ and $k'' = \arg\min_k \rho_k$. This is the first condition for the limit point of $V$ being an approximate KKT point. We can further show that $\chi \to 0$ as $t \to 0$ by noting the following alignment phenomenon which was originally observed by Ji & Telgarsky (2020) and intended for standard training with fixed training examples:

**Lemma 12** (Adapted from Theorem 4.1 in Ji & Telgarsky (2020)). *Under Assumption 2 and Assumption 4 for exponential loss and homogeneous deep neural networks, along the trajectory of adversarial training with scale invariant adversarial perturbations, we have*

$$\forall k \in \{1, \ldots, k\} : \lim_{t \to \infty} \frac{\left\langle W(t), \frac{\partial \tilde{\mathcal{L}}}{\partial W}(t) \right\rangle}{\|W\| \left\| \frac{\partial \tilde{\mathcal{L}}}{\partial W}(t) \right\|} = -1. \quad (132)$$

The extension of Theorem 4.1 in Ji & Telgarsky (2020), which was intended for fixed training examples, to our settings is because, by our construction for adversarial training with scale invariant adversarial perturbations, the adversarial training margin are locally Lipschitz with locally Lipschitz gradients and the prediction function $f(x + \delta(W); W)$ is positively (multi)homogeneous with respect to $W$ (see Lemma 10 and Eq. (4)). Invoking Lemma 12 and considering that $\hat{\gamma}_m$ can not be arbitrarily close to $0^8$ and $\forall k : \rho_k \to \infty$ at the same rate according to Lemma 11 thus $\frac{\rho_{k'}}{\rho_{k''}}$ can not be infinite, we have,

$$\lim_{t \to \infty} \chi(t) \to 0. \quad (133)$$

---

[8]See Eq. (105). In fact, it keeps increasing after some time according to Theorem 4.

On the other hand, the second condition for the limit point of $V$ being an approximate KKT point is

$$
\begin{aligned}
\sum_{i=1}^{n} \lambda_i(\frac{\tilde{\gamma}_i}{\tilde{\gamma}_m} - 1) = \sum_{i=1}^{n} \frac{\|W\|}{\tilde{\gamma}_m^{2/K}} \left( \left\| \frac{\partial \hat{\mathcal{L}}}{\partial W} \right\|_F \right)^{-1} e^{-\tilde{\gamma}_i}(\tilde{\gamma}_i - \tilde{\gamma}_m) \\
\overset{\mathsf{a}}{\leq} \frac{\sum_k \rho_k^2}{\rho^{2/K} \hat{\gamma}_m^{2/K}} \sum_{i=1}^{n} \frac{e^{-\tilde{\gamma}_i}(\tilde{\gamma}_i - \tilde{\gamma}_m)}{\left\langle W, \frac{\partial \tilde{\mathcal{L}}}{\partial W} \right\rangle} \\
\overset{\mathsf{b}}{\leq} \frac{\sum_k \rho_k^2}{\rho^{2/K} \hat{\gamma}_m^{2/K}} \sum_{i=1}^{n} \frac{n e^{-\tilde{\gamma}_i}(\tilde{\gamma}_i - \tilde{\gamma}_m)}{K \rho e^{-\tilde{\gamma}_m} \sum_j y_j f(x_j + \delta_j(\widehat{W}); \widehat{W})} \\
\overset{\mathsf{c}}{\leq} \frac{n \sum_k \rho_k^2}{K \rho^{1+2/K} \hat{\gamma}_m^{2/K}} \sum_{i=1}^{n} e^{-(\tilde{\gamma}_i - \tilde{\gamma}_m)} \frac{\tilde{\gamma}_i - \tilde{\gamma}_m}{\hat{\gamma}_m} \\
\overset{\mathsf{d}}{\leq} \frac{n \sum_k \rho_k^2}{K e \rho^{1+2/K} \hat{\gamma}_m^{1+2/K}} \\
\leq \frac{n L \rho_{k'}^2}{K e \rho \rho_{k''}^2 \hat{\gamma}_m^{1+2/K}} = \xi,
\end{aligned}
\tag{134}
$$

where $\mathsf{a}$ follows from $\left\langle W, \frac{\partial \tilde{\mathcal{L}}}{\partial W} \right\rangle \leq \|W\| \left\| \frac{\partial \tilde{\mathcal{L}}}{\partial W} \right\|$; $\mathsf{b}$ is due to Lemma 4 and $\forall i : \tilde{\gamma}_i \geq \tilde{\gamma}_m$; $\mathsf{c}$ uses $\forall i : \tilde{\gamma}_i \geq \tilde{\gamma}_m$ again; $\mathsf{d}$ is due to $e^{-x}x$ is upper bounded by its value at $x = 1$ for $x \geq 0$. Since $\forall k : \lim_{t \to \infty} \rho_k(t) \to \infty$ at the same rate, we conclude that $\lim_{t \to \infty} \frac{\rho_{k'}^2}{\rho_{k''}^2}$ can not be infinite thus $\lim_{t \to \infty} \xi(t) \to 0$ because $\hat{\gamma}_m$ can not be arbitrarily close to 0 and $\lim_{t \to \infty} \rho \to \infty$. Therefore, the limit point of $V$ is an $(\chi, \xi)$-approximate KKT point of the mini-norm problem along the trajectory of adversarial training with scale invariant adversarial perturbations where $\lim_{t \to \infty} \chi(t), \xi(t) \to 0$. Restating the theorem in Dutta et al. (2013) regarding the relation between $(\chi, \xi)$-approximate KKT point and KKT point in our setting:

**Theorem 6** (Theorem 3.6 in Dutta et al. (2013) and Theorem C.4 in Lyu & Li (2020))**.** *Let $\{x_t : t \in \mathbb{N}\}$ be a sequence of feasible point of the optimization problem 19. $x_t$ is an $(\chi_t, \xi_t)$-approximate KKT poiint for all $t$ with two sequences $\{\chi_t > 0 : t \in \mathbb{N}\}$ and $\{\xi_t > 0 : t \in \mathcal{N}\}$ and $\chi_t, \xi_t \to 0$. If $x_t \to x$ as $t \to \infty$ and MFCQ holds at $x$, then $x$ is a KKT point of this optimization problem.*

We then immediately conclude that the limit point of $\{W(t)/\|W\| : t \geq 0\}$ of gradient flow for the adversarial training objective Eq. (6) is along the direction of a KKT point of the optimization problem (19). $\square$

### B.4 GENERALIZATION TO NON-SMOOTH CASE

It is not hard to generalize the current analysis to non-smooth case, which will include the deep ReLU network, because, in our main steps, there are counterparts of our conclusions for non-smooth case for that sub-differential (Clarke, 1975) is a generalization of gradient. The non-smooth analysis can be done by first replacing the gradient flow equation Eq. (16) with its generalization,

$$
\frac{dW}{dt} \in -\partial \tilde{\mathcal{L}}(W(t))
\tag{135}
$$

where $\partial \tilde{\mathcal{L}}(W(t))$ is the sub-differential. Then one can follow similar procedures as our approach to make the generalization.

*Proof.* For simplicity, we only consider multi-1-homogeneous networks here, i.e., $c_1 = \cdots = c_L = 1$, which include the deep ReLU neural networks. Lemma 9 can be generalized to non-smooth case according to Lemma C.1 in Ji & Telgarsky (2020) or Theorem B.2 in Lyu & Li (2020):

**Lemma 13** (Lemma C.1 in Ji & Telgarsky (2020))**.** *Suppose $f : \mathbb{R}^n \to \mathbb{R}$ is locally Lipschitz and $L$-positively homogeneous, then for any $W \in \mathbb{R}^n$ and any $W^* \in \partial f(W)$,*

$$
\langle W, W^* \rangle = L f(W).
\tag{136}
$$

The starting point of the proof for Lemma 3 is the homogeneity of $f(x; W)$ and can be easily promoted to handle the non-smooth case.

With generalizations of Lemma 3 and Euler's theorem on homogeneous functions for non-smooth case, the proof of Lemma 4 can also be generalized to non-smooth case accordingly because it is based on the Euler's theorem on homogeneous functions and Lemma 3 for adversarial training. Specifically, it will become

$$\left\langle \frac{dW_k}{dt}, W_k \right\rangle = \frac{1}{n} \sum_{i=1}^{n} e^{-\tilde{\gamma}_i} \tilde{\gamma}_i. \tag{137}$$

The proof of Theorem 3 adopts Lemma 4 and can be generalized to non-smooth case by combining the chain rule (Clarke, 1983) for gradient flow to ensure Eq. (103)

$$-\frac{d\tilde{\mathcal{L}}}{dt} = -\left\langle \partial\tilde{\mathcal{L}}(W(t)), \frac{dW}{dt} \right\rangle = \left\| \frac{dW}{dt} \right\|_2 \geq 0 \tag{138}$$

with similar approach as in Lemma B.9 in Ji & Telgarsky (2020) or Lemma 5.2 in Davis et al. (2018). Then generalizing the rest of the proof of Theorem 3 is straightforward.

Finally, the generalization of Theorem 5 to non-smooth case can also be done because it adopts Theorem 3, Lemma 4, the chain rule, and the gradient alignments Lemma 12 which holds for non-smooth case. □

### B.5 DEEP LINEAR NETWORKS ADVERSARIALLY TRAINED WITH $\ell_\infty$ PERTURBATIONS

Following the settings of Section 3.1, the $\ell_\infty$-perturbation for deep linear networks $f(x; W) = W_L \cdots W_1 x$ can be given exactly as

$$\delta(W(t)) = -\epsilon y_i \text{sgn}(W_\Pi(t)). \tag{139}$$

According to Theorem 5, the adversarial training solution is formulated from solving the following optimization problem:

$$\max \frac{1}{2} \|W\|_2^2 \tag{140}$$

$$s.t. \min_{i \in \{1,\ldots,n\}} y_i W_\Pi x_i - \epsilon \|W_\Pi\|_1 \geq 1. \tag{141}$$

It can be seen that this optimization problem, a mixed-norm optimization problem, will have different solutions when compared to the margin-maximization problem of the original data, which will lead to a different decision boundary distinguishing adversarial training methods from the standard ones.

Besides, the constraint on $\epsilon$ (Eq. (8) for $\ell_2$ perturbation) should also be changed accordingly such that the adversarial training examples are still linear separable:

$$\epsilon \leq \gamma_{m,1} \tag{142}$$

where $\gamma_{m,1}$ is the max-$\ell_1$-norm margin:

$$\gamma_{m,1} = \max_{\|u\|_1=1} \min_{i \in \{1,\ldots,n\}} y_i u x_i. \tag{143}$$

## C SUPPLEMENTED EXPERIMENTS

### C.1 ADVERSARIAL TRAINING WITH DIFFERENT LOSS FUNCTIONS, ARCHITECTURES AND VARYING PERTURBATION SIZES

We present the results for adversarial training with different loss functions, architectures and varying perturbation sizes in this section to further verify our theorems and assumptions. We use two different models: one is a 3-layer neural network with the same architecture as that in the Section 4; the other is a CNN with architecture `Input-Conv-ReLU-Pooling-Conv-ReLU-Pooling-FC-FC`, where `Conv` stands for convolution layer, `FC` stands for fully connected layer and `Pooling` is max-pooling layer.

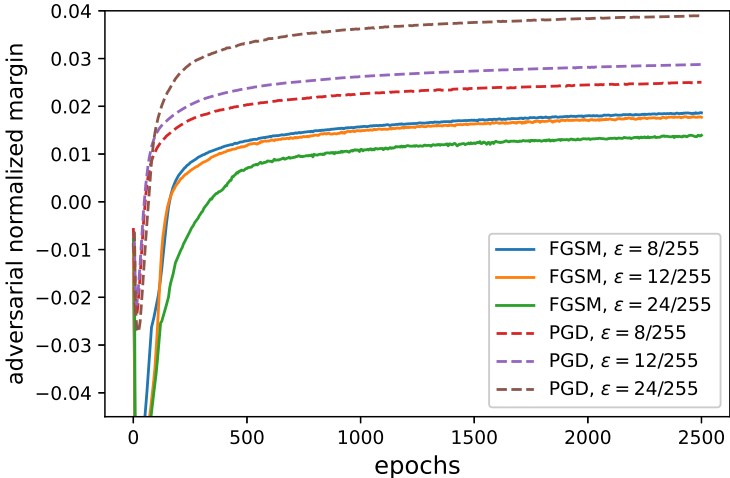

(a) Adversarial normalized margins for varying perturbation sizes

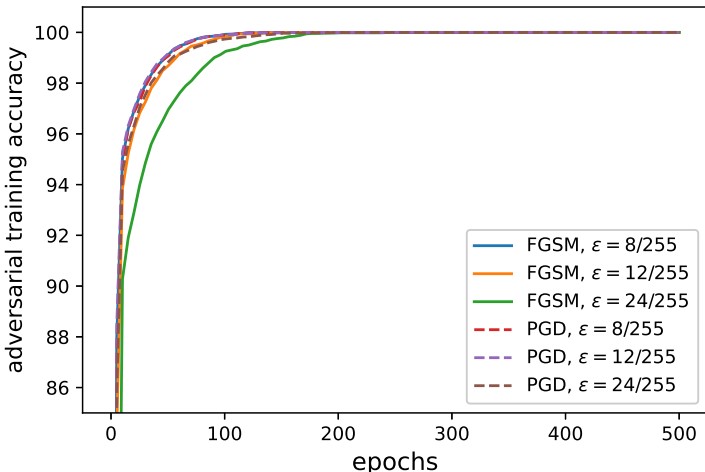

(b) Adversarial training accuracy for varying perturbation sizes

Figure 2: Adversarial training for the 3-layer neural network on MNIST to perform binary classification. The perturbations used for training are FGSM and $\ell_\infty$-PGD perturbations with varying perturbation sizes $\epsilon = 8/255, 12/255$ and $24/255$. The adversarial normalized margins and adversarial training accuracies are evaluated with the corresponding perturbations used for training.

For the 3-layer network, we conduct adversarial training using both FGSM and $\ell_\infty$-PGD perturbations with perturbation sizes $\epsilon = 8/255, 12/255$ and $24/255$ to perform binary classification where we choose all examples with labels "3" and "8" from MNIST to be our dataset. The loss function is logistic loss. Fig. 2(a) shows that for both FGSM and $\ell_\infty$-PGD perturbations with varying perturbation sizes, the adversarial normalized margins during training keep increasing after some step. Fig. 2(b) reveals that the adversarial training accuracy can achieve 100% (after $\sim 200$ epochs), which supports Assumption 4.

For the multi-classification task, let $f(x; W)[j]$ denote the $j$-th output. Then the margin for an adversarial example $x_i + \delta_i$ corresponding to the original example $x_i$ is defined by

$$\tilde{\gamma}_i = f(x_i + \delta_i; W)[y_i] - \max_{j \neq y_i} f(x_i + \delta_i; W)[j],$$

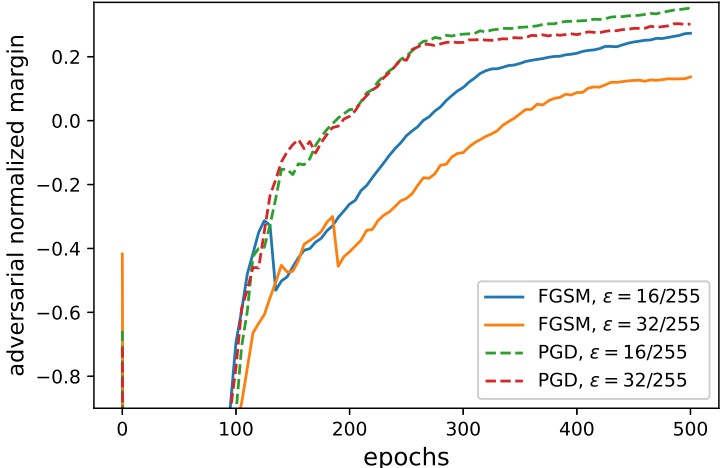

(a) Adversarial normalized margins for varying perturbation sizes

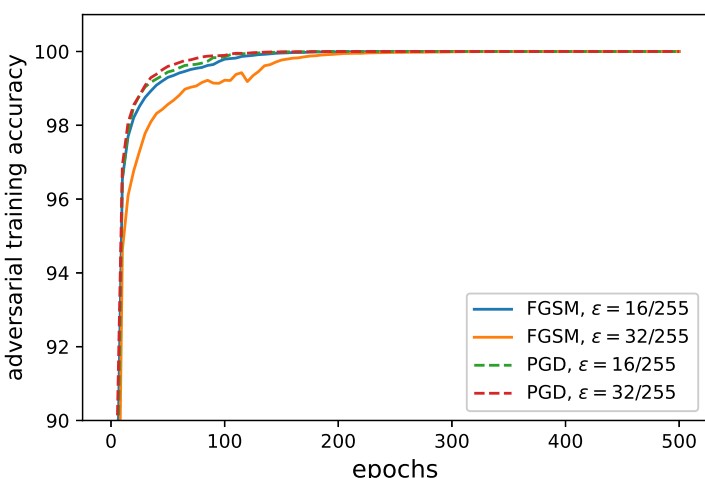

(b) Adversarial normalized margins for varying perturbation sizes

Figure 3: Adversarial training for the CNN on all training examples of MNIST to perform multi-classification. The perturbations used for training are FGSM and $\ell_\infty$-PGD perturbations with varying perturbation sizes $\epsilon = 16/255$ and $32/255$. The adversarial normalized margins and adversarial training accuracies are evaluated with the corresponding perturbations used for training.

and the margin for the adversarial data is defined to be

$$\tilde{\gamma}_m = \min_{i \in \{1, \dots, n\}} \tilde{\gamma}_i.$$

Then for the CNN, we conduct adversarial training using both FGSM and $\ell_\infty$-PGD perturbations with perturbation sizes $\epsilon = 16/255$ and $32/255$ to perform multi-classification on all training examples of MNIST to verify our claims and assumptions when the perturbation sizes are large. The loss function is cross-entropy loss. It can be seen from Fig. 3(a) that the adversarial normalized margins keep increasing after about 250 epochs, i.e., after the model fits all adversarial training examples, for all adversarial perturbations. Fig. 3(b) clearly shows that the adversarial training accuracy can reach 100% for all perturbation types and sizes, even the large size $\epsilon = 32/255$.

These experiments show that adversarial normalized margins are different whenever perturbation types or sizes are different. All models adversarially trained with all perturbations and loss functions can achieve 100% adversarial training accuracy, which verifies Assumption 4. Moreover, the trends that adversarial normalized margins all keep increasing are clear, even long after the separation of adversarial training examples. These results well support the claims of Theorem 5.

## C.2 EXPERIMENTS ABOUT $\ell_2$-FGM PERTURBATIONS FOR BINARY CLASSIFICATION

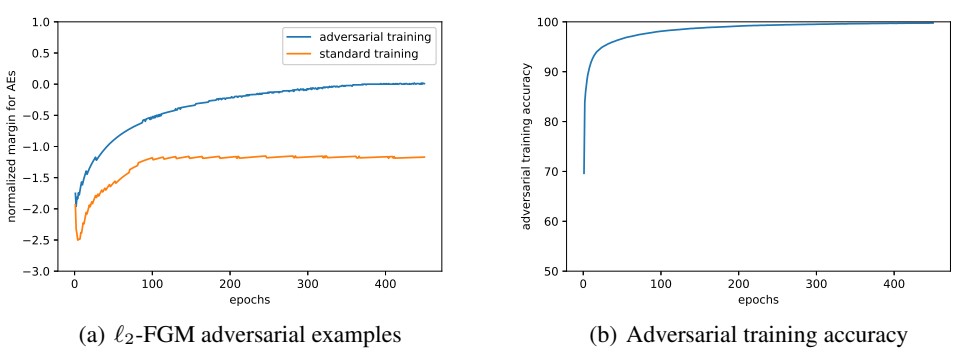

(a) $\ell_2$-FGM adversarial examples

(b) Adversarial training accuracy

Figure 4: Adversarial training for the 3-layer neural network on MNIST. The attack method is $\ell_2$-FGM with $\epsilon = 8$. (a) Adversarial normalized margin for $\ell_2$-FGM adversarial examples during adversarial training and standard training. (b) Adversarial training accuracy for $\ell_2$-FGM adversarial training.

In the MNIST dataset, we adversarially trained a 3-layer neural network with the same architecture as that in Section 4 using SGD with constant learning rate and batch-size 64. To have a clear view about the difference between the implicit bias for standard training and adversarial training, the adversarial perturbation used for training is $\ell_2$-FGM perturbation with $\epsilon = 8$ since even the standard model can fit $\ell_2$-FGM adversarial examples easily in the MNIST dataset when the perturbation size $\epsilon$ is not large. As a comparison, we also standardly trained a model with the same architecture to evaluate the normalized margin for adversarial examples by attacking it with $\ell_2$-FGM during its training process.

As showed in Fig.4(b), the adversarilly training accuracy reaches $100\%$ very quickly, which supports Assumption 4. It can be seen in Fig.4(a) that the adversarial normalized margin keeps increasing during the adversarial training process while the standardly trained model maintains a lower adversarial normalized margin.

### C.2.1 EXPERIMENTS ABOUT EXPONENTIAL LOSS

For completeness, we also conduct experiment about the binary classification where the model is adversarially trained with exponential loss (Fig.5) for $\ell_2$-FGM perturbations. We select the examples with label "2" and "8" from MNIST and adversarially trained a 2-layer neural network using SGD with constant learning rate $10^{-5}$. The hidden layer is of size 10000 and the activation function is ReLU. The normalized margin for adversarial data is defined exactly as in Section 3.2. The adversarial

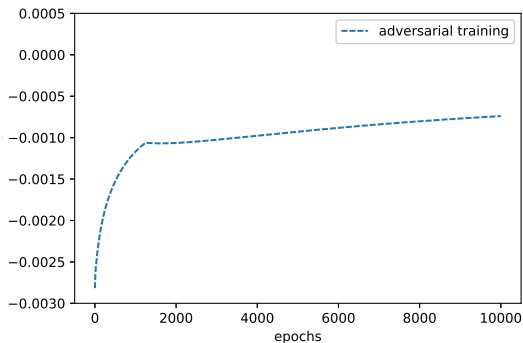

Figure 5: Adversarial normalized margin during adversarial training for binary classification. The loss function during training is taken as exponential loss.

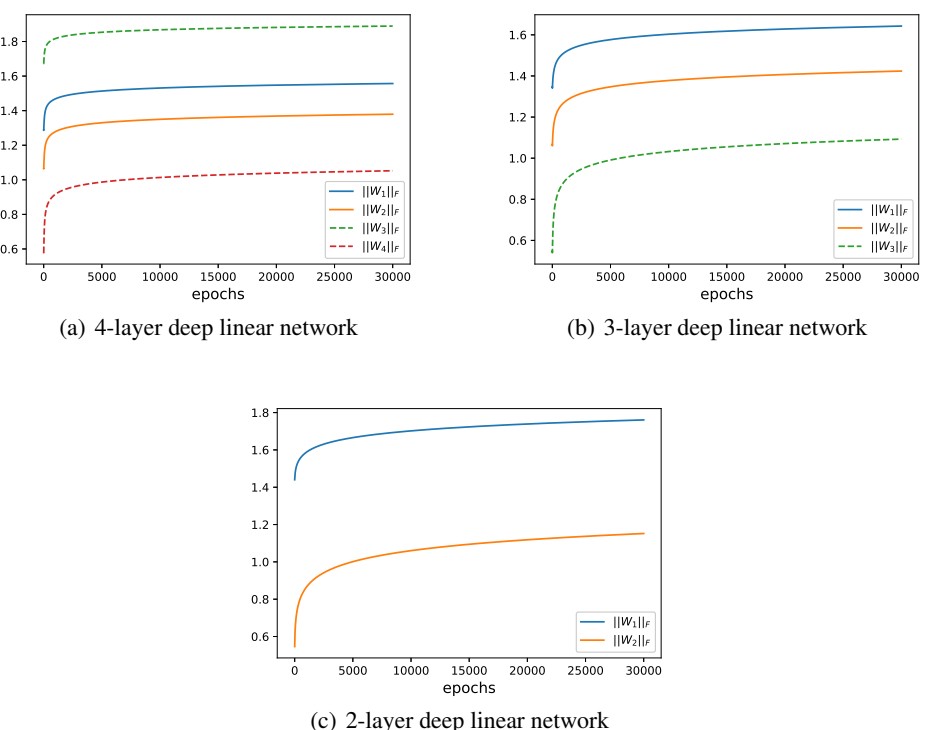

(a) 4-layer deep linear network        (b) 3-layer deep linear network

(c) 2-layer deep linear network

Figure 6: Frobenius norms for weights of deep linear networks with different layers during adversarial training. The perturbations are $\ell_2$ perturbations.

perturbations during the adversarial training are given by $\ell_2$-FGM with $\epsilon = 3$. As showed in Fig.5, although extremely slowly, the normalized margin for adversarial data gradually increases during the adversarial training.

### C.3 DIVERGENCE OF WEIGHT NORMS FOR DEEP LINEAR NETWORKS DURING ADVERSARIAL TRAINING

We conduct adversarial training of deep linear networks with $\ell_2$ perturbation on a linearly separable dataset. The loss function is logistic loss. Specifically, we use deep linear networks with layers 2, 3 and 4, respectively. Since the divergence is slow ($\sim \ln t$), it is hard to observe that $\|W_k\|_F \to \infty$.

However, as showed in Fig. 6(a), Fig. 6(b) and Fig. 6(c), the trends that $\|W_k\|_F$ for all layers of deep linear networks with different layers keep increasing are clear. Furthermore, we can see that the ratios $\|W_k\|_2/\|W_k\|_F$ for all layers of deep linear networks with different layers also keep increasing, as showed by Fig. 7(a), Fig. 7(b) and Fig. 7(c).

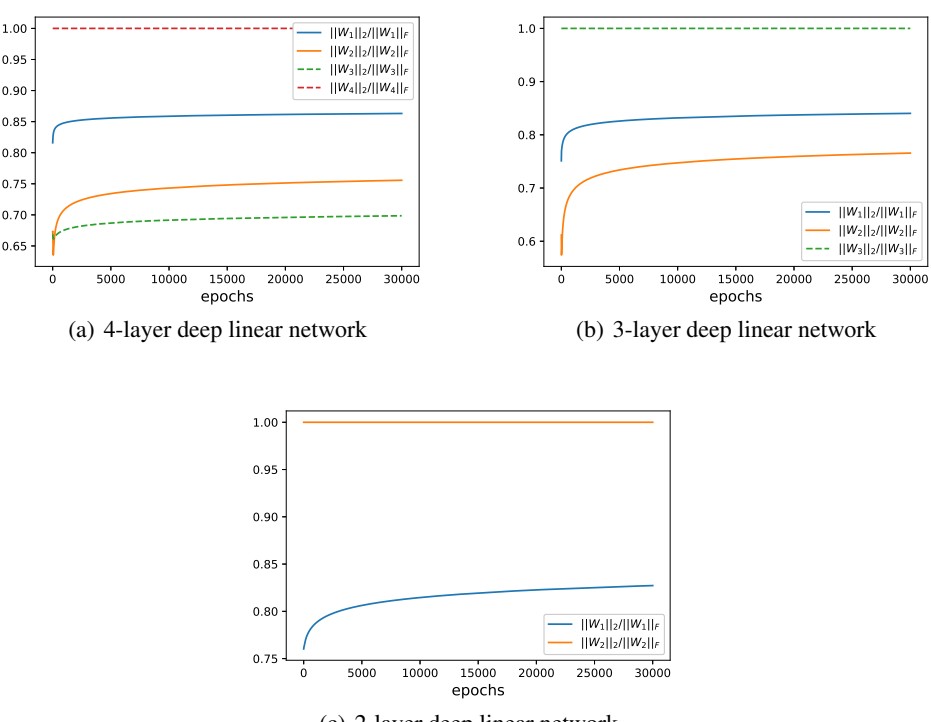

(a) 4-layer deep linear network

(b) 3-layer deep linear network

(c) 2-layer deep linear network

Figure 7: Ratios of 2-norm and Frobenius norm for weights of deep linear networks with different layers during adversarial training. The perturbations are $\ell_2$ perturbations.

