# OpenReview forum: "Implicit Bias of Adversarial Training for Deep Neural Networks"
_ICLR.cc/2022/Conference — ICLR 2022 Poster_

### Official Review · Reviewer_kpiH · 2021-10-30

**Correctness:** 4
**Technical Novelty And Significance:** 3
**Empirical Novelty And Significance:** 4
**Recommendation:** 8
**Confidence:** 3

**Main Review:**

Some minor concerns:
1. the notation of loss function is abused. The loss function in (6) takes both x and y as the input, while in Assumption 1, the loss function only takes one argument. I suppose the loss function in (6) means l(x,y;W) = l' (f(x;W)*y) where l' is the loss function in Assumption 1.
2. I don't quite understand the remark at the top of page 9. LHS of (20) is about the prediction of the clean test sample, the RHS of (20) is about the fitting of the adversarial training sample. How does this inequality relate to the trade-off between robustness and accuracy? Please add more discussion.
3. Page 21, after eq (81), "and Lemma 4 can be applied to ". It should be "and Lemma 9 can be applied to".

**Summary Of The Paper:**

This paper aims to understand the training results of adversarial training, and proves that under certain conditions, adversarial training results maximize the margin for the adversarial training samples. Similar results have been observed in the cleaning training of DNN, this paper's contribution is to extend them to the adversarial training settings. The paper seems technically sound (although I don't have the time to go over all the appendix). The results, to be honest, are not surprising, given previous works on standard training. But I believe the rigorous justification presented in the paper is of importance.

**Summary Of The Review:**

The theoretical contribution of this paper is good to me (unless other reviewers find technical errors in the proof). I think it is a good supplement to the current theory of adversarial DNN training.

---

> ### Author Response · Authors · 2021-11-19
> **Thanks for your reviews.**
>
> Thanks a lot for appreciating our work and your suggestions.
> - We fixed the notation of loss function in Eq.~(6) in the revision as $\ell(y_if(x_i;W))$.
>
> - The inequality in Eq. (20) implies that the adversarially trained model $f(\cdot; W)$ will classify any clean examples $x' = x + \tau$ around $x$ as having labels 1, even though there may be some of them which have true labels $-1$. In other words, to correctly classify the adversarial example $x + \delta(W)$, our model $f(\cdot; W)$ can not correctly classify some clean examples. Thus we have the trade-off between robustness and accuracy.
>
> - Thanks for pointing this out. We fixed this problem in the revision.

---

### Official Review · Reviewer_eZ8u · 2021-11-03

**Correctness:** 3
**Technical Novelty And Significance:** 3
**Empirical Novelty And Significance:** Not applicable
**Recommendation:** 5
**Confidence:** 3

**Main Review:**

This paper focuses on the convergence behavior of adversarial training methods. The paper tries to show the implicit bias of adversarial training in a simplified setting with a deep linear neural network and linearly separable data for a binary classification problem. Under this setting, the paper proves the gradient descent algorithm will converge asymptotically to the max-margin solution. Later, the paper extends this result to homogeneous neural network functions with the exponential loss function and a separation assumption in Assumption 4. Overall, the paper targets an interesting question and proves some useful results on the behavior of adversarial training problems. However, I have some comments on the assumptions made to simplify the analysis, some of which seem to be quite restrictive and limit the results' application to real adversarial training problems.

Regarding the paper's theoretical setting, I think some of the assumptions are quite strong. In section 3.1, the analysis is limited to deep linear networks and linearly separable data in binary classification. Also, the convergence result is an asymptotic guarantee which does not bound the iteration complexity of finding the max-margin solution. The convergence guarantee to the max-margin solution also holds for the standard training algorithm, which questions whether the result can distinguish adversarial training methods from standard training algorithms.

While the results in section 3.1 are written clearly, I think section 3.2 lacks a clear presentation and puts several limiting assumptions. First, the gradient steps are replaced with gradient flow which is inconsistent with real adversarial training experiments. Also, the exponential loss function used for theoretical analysis is not used in practical adversarial training experiments. It is not clear whether the results can be extended to standard cross-entropy and squared-error loss functions in deep learning classification problems. Also, Assumption 4 on the separability of adversarial examples is pretty strong and essentially assumes the adversarial training method finds a perfect solution at some iteration t_0, which is too strong given that the paper wants to study the convergence behavior of adversarial training. Therefore, I think the assumptions are too restrictive for a real adversarial learning setting and significantly limit the application of the results to practical deep learning experiments. I will look forward to the authors' responses regarding the reasoning behind these assumptions to give my final score.

**Summary Of The Paper:**

The paper studies the adversarial training problem under deep linear network classifiers and standard L_2 and L_\infty perturbations. The paper's main result suggests that in the linearly separable case the adversarially trained model via gradient descent will asymptotically converge to the max-margin solution. Some extensions of this result to homogenous neural networks with exponential loss function have been provided. The paper also performs some preliminary numerical experiments to support the theoretical results.

**Summary Of The Review:**

While the paper shows some insightful results on the convergence of adversarial training for deep linear networks, the assumptions for the analysis of deep nonlinear networks seem too restrictive to me. Also, replacing the gradient steps with the gradient flow seems incompatible with standard adversarial training experiments. The paper would become much stronger after relaxing some of the assumptions and performing the analysis for the actual gradient descent algorithm rather than considering the gradient flow.

---

> ### Author Response · Authors · 2021-11-19
> **Thanks for your reviews.**
>
> Thanks a lot for your view. Regarding your concerns, we have the following response.
> 1. **Iteration complexity.** To the best of our knowledge, currently there is even no work to derive a clear iteration complexity bound for the standard training of deep linear networks. In the current paper, our focus is the implicit bias of adversarial training for deep neural networks. We believe future work can explore the iteration complexity to compare adversarial training and standard training based on our results. Besides, by applying Theorem 5 to deep linear networks (please see the Appendix B.5 in the revision), one can get that adversarial training gives solutions formulated from solving a mixed-norm margin maximization problem, which is able to reveal the difference between adversarial training and standard training. We acknowledge that our novel  contribution is only a starting point for fully understanding the implicit bias of adversarial training. More comprehensive analysis, including iteration complexity, will be left as future work.
>
> 2. **Loss functions.** We adversarially trained a 3-layer network with logistic loss to perform binary classification (please see Fig.2 in the Appendix C.1 of the revision) and a CNN on MNIST with cross-entropy loss to perform multi-classification (please see Fig.3 in the Appendix C.1 of the revision). The activations for both nets are ReLU. The results show that the adversarial normalized margins keep increasing for both logistic loss and cross-entropy losses. Since both logistic loss and cross-entropy loss are exponential-type losses, the results are as expected.
>
> 3. **Gradient flow.**  We have the following reasons for using gradient flow in the current work. First, gradient flow for adversarial training can be seen as gradient descent with infinitesimal step size. It is a widely-used approach to model gradient descent to simplify the analysis, e.g., Wei et al., 2019 and Ji \& Telgarsky, 2020. Second, our work takes the first step towards the theoretical analysis of the implicit bias of adversarial training. Our results can provide significant insights to the robustness research community by using gradient flow, as also suggested by other reviewers. These results are also further verified by different practical experiments. Third, the results of gradient flow can largely inspire the analysis of gradient descent by following similar approach as that of gradient flow.
>
> 4. **Assumption 4.**  We would like to highlight that achieving 100\% adversarial training accuracy, i.e., fitting all adversarial training examples, is not essentially equivalent to finding perfect solution (it is also weaker than assuming zero adversarial training loss). We supplemented many experiments to verify this assumption: we adversarially trained 3-layers ReLU nets (please see Fig.2(b) in the Appendix C.1 of the revision) and CNN (please see Fig.3(b) in the Appendix C.1 of the revision) with varying perturbation sizes and loss functions whose results show that adversarial training accuracies achieve 100\% in all cases. In fact, the training dynamics continue after the separation of adversarial training examples. For an example, the adversarial normalized margin keeps increasing long after the separation of adversarial training examples (Fig.1(c)). Therefore, assuming separability of adversarial training examples to study the implicit bias is meaningful in the current work.

---

### Official Review · Reviewer_QB5M · 2021-11-03

**Correctness:** 4
**Technical Novelty And Significance:** 4
**Empirical Novelty And Significance:** Not applicable
**Recommendation:** 8
**Confidence:** 3

**Main Review:**

Strengths:
- The results are novel and extend prior theoretical results.
- To the extent I have verified, the proofs are correct.


Minor comments:
- Theorem 5: one limitation of this result is that it depends on the adversarial perturbation as part of the constraints in Eq 19. That is in comparison with related results of Li et al 2020 and Faghri et al 2021 that their characterizations make the difference between the solution for various Lp-norm perturbations clear. Understandably, Theorem 5 is a more general result for homogenous models but it would still be useful to derive prior linear results as corollaries of Theorem 5.
- Assumption 4 can easily be false for large perturbation sizes. The footnote says similar assumptions have been made in prior work but those were not about separability of adversarial examples. Can you provide more justification for this assumption?
- Figure 1: This is an interesting plot confirming the increase in adversarial margin. Can you plot FGSM and PGD on the same plot? I understand that the adversarial margin for the two is different because the corresponding optimization problem is different. However, a natural question is, is there a relation between the two problems?
- Page 9, Trade-off between standard and adversarial accuracy: I’m not sure I understand the theoretical argument of this part. Is there a concrete result based on Theorem 5?
- Section 4: Have you verified these results for varying epsilon size (other than 16/255)? How about other network architectures? Is there a challenge in doing so?


**Summary Of The Paper:**

This paper characterizes the bias of adversarial training toward specific minimum-norm solutions or KKT points of a particular optimization problem. Their results generalizes the work of Li et al 2020 by proving the directional alignment with the adversarial max-margin solution for deep linear models for L2 perturbations (Theorem 2) as well as convergence in direction for homogenous networks for L2 FGM, FGSM, and L2, Linf PGD perturbations (Theorem 5).

**Summary Of The Review:**

This paper makes a solid theoretical contribution. It could be improved with more empirical verification of the results.

---

> ### Author Response · Authors · 2021-11-19
> **Thanks for your positive reviews. We made improvements in the revision based on your suggestions**
>
> Thanks a lot for your positive review and suggestions. We made the following improvements in the revision based on your suggestions.
>
> 1. We added discussion on applying Theorem 5 to derive the results for adversarial training of deep linear networks with $\ell_{\infty}$ perturbations in Appendix B.5. Specifically, for the deep linear network with $\ell_\infty$ perturbation, the adversarial training solution is formulated from solving the following optimization problem:
>   \begin{equation}
>     \min \frac12 ||W||_2^2
>   \end{equation}
>   \begin{equation}
>     s.t. \min_\{i\in\{1,\dots,n \}} y_iW_\Pi x_i - \epsilon ||W_\Pi ||_1 \geq 1.
>    \end{equation}
> This optimization problem, a mixed-norm optimization problem, will have different solutions when compared to the margin-maximization problem of the original data. Therefore we will have a different decision boundary when compared to that of the solution returned by standard training or adversarial training with $\ell_2$-perturbations. This also clearly reveals the difference between adversarial training and standard training for deep linear networks. By taking the number of layers $L$ as 1, one can recover the results in Li et al, 2020 for $\ell_\infty$-perturbation.
>
> 2. From a theoretical perspective, Zhang et al, 2020 and Gao et al, 2020 proved the convergence to low adversarial training error for overparametrized models. Since Assumption 4 only requires separability of adversarial examples during training (this is weaker than achieving low or zero training loss), these theoretical results can justify it.
>
>       From the empirical perspective, overparametrized DNNs can usually achieve 100\% adversarial training accuracy in practice, e.g., Table 1 in Raghunathan et al., 2019. In fact, we rarely stop training immediately when our models achieve 100\% training accuracy in practice. Making Assumption 4 is acceptable in this sense. Besides, we also adversarially trained a DNN with varying perturbation sizes ($\epsilon =  8/255, 12/255$ and $24/255$) and plotted the adversarial training accuracy (please see Fig.2(b) in the Appendix C.1 of the revision for details). It can be seen that all models adversarially trained with different perturbation sizes can achieve 100\% training accuracy, i.e., separation of adversarial examples during training. These results can verify Assumption 4 for overparametrized DNNs with different perturbation sizes in a certain range.
>
> 3. Please see Fig.2(a) and Fig.3(a) in Appendix C.1 of the revision for details, where we plot results for FGSM and PGD perturbations with varying perturbation sizes in the same plot. The results show that the adversarial normalized margins are different when either the perturbation sizes or perturbation types are different while the trends that adversarial normalized margins keep increasing are clear which further verifies Theorem 5.
>
> 4. Following the discussion in Page 9 regarding the trade-off between robustness and accuracy, as a consequence of forcing our model to correctly classify the adversarial example $x + \delta(W)$, $f(\cdot; W)$ will classify any clean test examples $x' = x + \tau$ around $x$ as having labels $y' = 1$ according to Eq.~(20), even though there may be some of them which have true labels -1. This is an inevitable trade-off when increasing robustness of our model through adversarial training. Therefore we have the tension between robustness and accuracy.
>
>     We believe that deriving concrete results for the trade-off between robustness and accuracy (e.g., a quantity clearly bounding this trade-off) based on Theorem 5 is an intriguing direction and we leave this for future work.
>
> 5. We conducted experiments for adversarial training with varying perturbation sizes (8/255, 12/255 and 24/255) for a 3-layer-networks. We also adversarially trained a CNN using cross-entropy loss to perform multi-classification on MNIST with FGSM perturbations and $\ell_\infty$ PGD perturbations (the perturbation sizes are 16/255 and 32/255).  It can be seen that in all experiments: 1) the adversarial normalized margins keep increasing after some step, which means that for this CNN architecture and other loss functions the claim of Theorem 5 also holds; 2) the adversarial training accuracies all achieve 100\% after some step, which verifies that making Assumption 4 is reasonable.
>
>     Please see Fig.2 and Fig.3 in the Appendix C.1 of the revision for these supplemented experiments.
>
> ---
> Reference:
>
> Over-parameterized adversarial training: An analysis overcoming the curse of dimensionality. Zhang et al., 2020.
>
> Convergence of adversarial training in overparametrized neural networks. Gao et al., 2019.
>
> Adversarial Training Can Hurt Generalization. Raghunathan et al., 2019.

---

> > ### Comment · Reviewer_QB5M · 2021-11-22
> > **Thank you for your response**
> >
> > I thank the authors for their responses and improvements to the paper. I encourage the authors to incorporate the remainder of details from their responses into the paper. I keep my recommendation for acceptance.

---

### Official Review · Reviewer_uXwk · 2021-11-08

**Correctness:** 2
**Technical Novelty And Significance:** 4
**Empirical Novelty And Significance:** 4
**Recommendation:** 5
**Confidence:** 4

**Main Review:**

*Implicit Bias of Adversarial Training for Deep Neural Networks*
answers a prevalent question in the theory of adversarial robustness:
how exactly (in closed mathematical form) does adversarial training
improve adversarial robustness? The paper proves a theorem stating
adversarial training produces an implicit bias on the normalized
weights $\frac{W}{\lVert W \rVert}$ (with a more precise statement in
Summary Of The Paper). The majority of the paper is clearly written
and correct. The paper's results place a milestone in the theory of
adversarial robustness. However, the proof of Theorem 2 may have
potential errors (which may be from my confusion on some statements
and notation). Adversarial training in linear neural networks is a corollary to Theorem 5,
so any potential errors in Theorem 2 do not invalidate Theorem 5's results.
 In addition, there are several statements and notation
that are vague and ill-defined in the theorems. This makes the exact
statement of the theorems difficult to ascertain.

## Potential errors in the proof of Theorem 2

Lemma 5 requires a constant step size $1/\beta(r, \bar{r}, \epsilon)$
for
$$ \beta(r, \bar{r}, \epsilon) = r^{3L}L^2\left[ \alpha + \beta +
\frac{\epsilon}{\bar{r}^L} \left( 2\alpha + \beta + frac{\alpha
\beta}{\bar{r}^L} \right) \right].$$
The constant step size implies in Lemma 5 that
$$\max_k \lVert W_k(t) \rVert > r(t)$$
for some $t$. However, in Lemma 2, the step size $\eta(t)$ is not constant where
$$\eta(t) = \min \{ 1, \beta(r(t), \bar{r}(t), \epsilon) \}$$
and
$$r(t+1) = r(t) + \mu(t)$$
for $W(t+1) \not\in \mathcal{S}(r(t) - \mu(t))$. As the $\eta(t)$ decreases whenever
$W(t+1) \not\in \mathcal{S}(r(t) - \mu(t))$, the step size $\eta(t)$
decreases at $\mathcal{S}(r(t) - \mu(t))$ and not at $\mathcal{S}(r(t))$.
It is not obvious that Lemma 5 holds as the step size is not constant in $\mathcal{S}(r(t))$.
This is particularly important as a Lemma 6 and Theorem 2 assume that
 $\max_k \lVert W_k \rVert_F \to \infty$. Could you please provide a short proof
that $\max_k \lVert W_k \rVert_F \to \infty$ under the assumptions of Theorem 2?

It is also unclear how statements under
Equation 39 hold as a result of $\max_k \lVert W_k \rVert_F \to \infty$. Why does
$$U_k \Sigma_k \Sigma_k^T U_k^T \to V_{k+1}\Sigma^T_{k+1}\Sigma_{k+1}V^T_{k+1}?$$
Why does this imply $\Sigma_k \Sigma^T_k$ and $\Sigma^T_{k+1}\Sigma_{k+1}$ are
"approximately the same"? How is "approximately the same" defined? Why do all layers
have rank 1?

## Vague language, ill-defined statements, and undefined definitions

The paper suffers from several instances of vague language. For
instance, "alignment phenomenon" on pages 6, 15, and 17 are not
defined in the paper. Although I can figure out your intended
definition, it makes Theorem 2's precise statement difficult to
ascertain.  Other instances include "approximately the same" on page
15 and improper use of limits on Equation 52.  In addition, the
assumptions in the theorems sometimes do not match the proofs.  The
proof of Theorem 2 assumes only a logistic loss function while Theorem
2's statement assumes a broader class of loss functions.

## Limited experimental results

**These comments did not impact my review recommendation.**

In many theoretical papers, experimental sections typically
quantitatively measure the difference between general theoretical
statements and common real-world cases. For example, experiments show
how theorems in compressed sensing deviated from typical use cases. In
the paper, a particular area for improvement is the experimental
section. This section show the training accuracy and the normalized
margin. Both plots will clearly increase as a result of adversarial
training and do not add any useful information to the paper. Do the
weights' singular values diverge to infinity in practice? Does Theorem
5's statement on the normalized weights still occur when you use other
losses and neural networks or remove Assumptions 2 and 4?


**Summary Of The Paper:**

*Implicit Bias of Adversarial Training for Deep Neural Networks*
explores how minimizing the exponential loss (i.e., $l(x) = e^{-x}$)
of a homogeneous neural network (i.e., a neural network such that $$
f = a_L W_L \circ \sigma_L \circ \cdots \circ \sigma_2 \circ a_1 W_1 =
\prod^L_{k=1} a_k^c (W_L \circ \sigma_L \circ \cdots \circ \sigma_2 \circ
W_1 $$ for activation functions $\sigma_L, \ldots, \sigma_2$, weights
$W_L, \ldots, W_1$, $a_L, \ldots, a_1 > 0$, and $c \geq 1$) on samples
with perturbations that maximizing the loss influences the optimized
neural network's weights. Specifically, this paper proves that, for an
exponential loss and a multi-c-homogeneous neural network, the limit point for $\frac{W}{\lVert W \rVert}$
with respect to the gradient flow
$$
\frac{dW}{dt} = - \left( \frac{d\tilde{\mathcal{L}}}{\partial W} \right)^T
$$
of the adversarial training objective
$$
\tilde{\mathcal{L}} = \frac{1}{n}\sum^n_{i=1}\ell(x_i + \delta_i(W), y_i)
$$
under $\ell_2$-FGM, FGSM, $\ell_2$-PGD, and $\ell_\infty$-PGD is along the Karush-Kuhn-Tucker (KKT) point
of the constrained minimization problem
$$
\min_{W_1, \ldots, W_L} \frac{1}{2} \lVert W \rVert^2_{\ell_2} \text{ s.t. } \tilde{\gamma_i} \geq 1
$$
where $\tilde{\gamma_i} = y_i f(x_i + \delta_i(W))$ and $W = (W_1, \ldots, W_L)$.

This theorem demonstrates that---for a class of neural networks and adversarial perturbations---adversarial training has
an implicit bias that can be expressed in closed form. This result provides an important contribution to understanding
how adversarial training improves adversarial robustness.


**Summary Of The Review:**

*Implicit Bias of Adversarial Training for Deep Neural Networks*
contributes a milestone result to the theory of adversarial
robustness. The majority of the paper is clearly written and
correct. However, the Adversarial Training for Linear Neural Networks
portion of the paper has several flaws: potential errors, vague
language and ill-defined statements (such as "approximately the same"
and improper use of limits in Equation 52), and use of undefined
notation and definitions (such as alignment phenomenon). Nevertheless,
the contributions are significant and novel, and the paper would
receive an accept if Adversarial Training for Linear Neural Networks
is amended or removed.

---

> ### Author Response · Authors · 2021-11-19
> **Thanks for your reviews. Our response Part 1/2**
>
> Thanks a lot for your review and suggestions. Below we address your concerns.
> - **Regarding Potential errors in the proof of Theorem 2.**
>     1. **On $\max_k ||W_k||_F \to \infty$.** We would like to clarify that we do not directly apply Lemma 5 at $S(r(t))$ to prove $\max_k ||W_k||_F \to \infty$. Instead, we repeatedly apply Lemma 5 when it holds, i.e., at $S(r(t) - \mu(t))$, during the adversarial training under assumptions of Theorem 2. Specifically, for any given $r(t_1)$, we first perform gradient descent with constant $r(t_1)$ and $\eta(t_1)$ thus conditions of Lemma 5 hold and we can apply Lemma 5 to conclude that there will be a time $t_2 > t_1$ such that $W(t_2 + 1) \notin S(r(t_2) - \mu(t_2)) = S(r(t_1) - \mu(t_1))$. After $t_2$, we let $r(t_2 + 1) = r(t_2) + \mu(t_2) = r(t_1) + \mu(t_1)$ and change $\eta(t_2 + 1)$ accordingly. Then we keep them as constants thus Lemma 5 also holds. By repeatedly applying Lemma 5 in this manner, we can conclude that, for any given $R(t) = r(t) - \mu(t)$, there will be a time such that $\max_k ||W_k||_F > R(t)$ thus $\max_k ||W_k||_F \to \infty$ is unbounded.
>     2. **Statements under Eq. (39).** The statements under Eq. (39) are established in the approximate sense. We first answer your questions. According to Eq. (36),  since $||W_k||_F \to \infty$ as $t \to \infty$, the initializations of them (constant matrices) and $A_\{k\}, B_\{k+1\}$ (traces upper bounded by Eq. (38)) become negligible thus Eq. (36) implies that $W^T_k(t)W_k(t) \to W_\{k + 1\}(t)W^T_\{k + 1\}(t)$. Then $U_k\Sigma_k\Sigma_k^TU_k^T$ and $V_\{k + 1\}\Sigma_\{k + 1\}^T\Sigma_\{k + 1\}V_\{k + 1\}^T$ are singular value decompositions for "approximately" the same matrix thus $\Sigma_k\Sigma_k^T$ and $\Sigma_\{k + 1\}^T\Sigma_\{k + 1\}$ are “approximately the same”. If $k + 1 =L$, we know that $W_\{L\}W^T_L$ has rank 1 because $W_L$ is simply a vector. Since any two adjacent layers get aligned, layers other than $W_L$ will also “approximately” have rank 1 thus $W_k / ||W_k||_F \to  u_kv_\{k\}^T$.
>
>          To further elaborate the above reasoning and fix the vague language (and undefined definitions) problem in your next concern, we first define the alignment phenomenon formally (see the following definition). With this definition, we then provide a more exact proof for the “alignment phenomenon” in the Section A.2.1 of the revision and present the proof sketch here. The proof basically shows that $W_k$’s left singular vector, $u_{k}$, and $W_{k + 1}$’s right singular vector, $v_{k + 1}$, corresponding to their largest singular values are the same as a result of $\sigma_k \to \infty$. Furthermore, $W_k$ and $W_{k + 1}$ share the same largest singular value, i.e., $\sigma_k \to \sigma_{k + 1}$, and other singular values of $W_k$ are negligible, i.e., $\forall k: ||W_k||_2 / ||W_k||_F \to 1$, as a result of $\sigma_k \to \infty$. Therefore, we have that $W_k / ||W_k||_F \to u_kv_k^T$ and any adjacent layers get aligned. These results will clearly reveal the alignment phenomenon and formally justify the statements under Eq. (39). We begin with the definition.
>
>         **Definition (Alignment phenomenon of adversarially trained deep linear networks).** For deep linear network $f(x; W) = W_L\cdots W_1x,$ the alignment phenomenon is defined as
> \begin{equation}
>     \forall k \in \{1, \dots, k \}: |\left<u_k, v_{k + 1}\right> | \to 1
> \end{equation}
> and $W_k / \|W_k\|_F \to u_kv_k^T$ as $t \to \infty$ along the training trajectory, where $u_k, v_k$ are the left and right singular vectors correspond to the largest singular value of $W_k$, $\sigma_k$.
>
>         We now present the proof sketch for the alignment phenomenon.
>
>          *Proof.*  **Step 1.** Note that
> \begin{equation}
>         \left<u_k, v_{k + 1}\right>^2\sigma_{k + 1}^2 =  u_k^TW_{k + 1}^TW_{k + 1}u_k + u_k^T(v_{k + 1}\sigma_{k + 1}^2v_{k + 1}^T -  W_{k + 1}^TW_{k + 1})u_k,
> \end{equation}
> we have \begin{equation}
>      \frac{u_k^TW_\{k + 1\}^TW_\{k + 1\}u_k}{\sigma_\{k + 1\}^2} + \frac{||W_\{k + 1\}||_2^2 - ||W_\{k + 1\}||_F^2}{\sigma_\{k + 1\}^2} \leq\left<u_k, v_\{k + 1\}\right>^2 \leq 1
> \end{equation}
> by utilizing the definition of matrix norm and
> \begin{equation}
>     ||W_\{k + 1\}||_F^2 \geq u_k^TW_\{ k + 1\}^TW_\{k + 1\}u_k.
> \end{equation}
>
>          **Step 2.**  According to Eq. (36), let
> \begin{equation}
>     \Gamma_k = W_\{k + 1\}^T(0)W_\{k + 1\}(0) - W_k(0)W_k(0)^T + A_\{k + 1\} -B_k,
> \end{equation}
> then we have, from the results in the Step 1,
> \begin{equation}
>     \left<u_k, v_\{k + 1\}\right>^2 \geq \frac{\sigma_k^2}{\sigma_\{k + 1\}^2} + \frac{u_k^T\Gamma_ku_k + ||W_\{k + 1\}||_2 - ||W_k||_F^2}{\sigma_\{k + 1\}^2}.
> \end{equation}
>
>         **Step 3.**   Please see Part 2 of our response for Step 3.

---

> > ### Author Response · Authors · 2021-11-19
> > **Our response Part 2/2**
> >
> > - **Step 3.**  To prove the alignment phenomenon, we show that the RHS of the above equation are lower bounded by $1 - \alpha$ with $\alpha \to 0$ when $t \to \infty$ as a result of $\sigma_k \to \infty$, which implies that $\left<u_k, v_{k + 1}\right>^2 \to 1$. Furthermore, since $||W_\{k + 1\}||_F^2 - ||W_\{k + 1\}||_2^2$ is finite and $\sigma_k \to \infty$ (i.e., $||W_k||_2^2 / ||W_k||_F^2 \to 1$), we have $\frac{W_k}{||W_k||_F} \to u_kv_k^T$ because other singular values are negligible. The details are presented in the Appendix A.2.1 in the revision. QED.
> >
> > - **Regarding Vague language, ill-defined statements, and undefined definitions.** Please see the point 2 of our response regarding the potential errors in the proof of Theorem 2, where we give the definition of the alignment phenomenon and a new exact proof to prove it which fixes the vague language problem.
> >
> >     Thanks for pointing the statement regarding loss function in Theorem 2. We fixed this in the revision.
> >
> > - **Regarding limited experimental results.**
> >
> >    1. We conducted the adversarial training of a 3-layer linear network on a linearly separable dataset. Since the convergence rate is very slow ($\sim \ln(t)$ according to Theorem 3), the divergence of singular values is hard to observe but the trend that $||W_k||_F$ keeps increasing  and that $||W_k||_2 / ||W_k||_F \to 1$ (i.e., singular values other than the largest one are negligible) are observed clearly. Please see Fig.6 and Fig.7 in the Appendix C.3 of the revision for details.
> >
> >    2. For other losses and neural networks, we supplemented  experiments where we adversarially trained a CNN with ReLU activation and max-pooling layers, which makes it a homogeneous neural network, on MNIST to perform multi-classification problem with cross-entropy loss. We used four perturbations, namely FGSM with $\epsilon = 16/255$, FGSM with $\epsilon = 32/255$, $\ell_\infty$-PGD with $\epsilon=16/255$, and  $\ell_\infty$-PGD with $\epsilon=32/255$. The results show that in all experiments: 1) the adversarial normalized margins keep increasing after some time $t_0$ during training; 2) the adversarial training accuracies finally reach 100\%, even for the large perturbation size $32/255$. Please see Fig.3(a) and Fig.3(b) in the Appendix C.1 of the revision for details. These experiments can further verify the theoretical claims of Theorem 5 and that Assumption 4 can be satisfied in practice even for large perturbation sizes and other loss functions.
> >
> >
> >        Besides, we also adversarially trained a 3-layer neural network with the same architecture as in Section 4 using varying perturbation sizes and logistic loss to classify examples with labels $3$ and $8$. Please see Fig.2(a) and Fig.2(b) in the Appendix C.1 of the revision for details. These experiments also verify Theorem 5 where adversarial normalized margins all keep increasing after the separation of adversarial training examples. And adversarial training accuracies in all experiments achieve $100\%$, which supports that making Assumption 4 is reasonable in practice.

---

### Author Response · Authors · 2021-11-29
**To all reviewers**

We thank all reviewers for their valuable comments and suggestions. In the revision, we made the following improvements to address the raised concerns:
- We supplemented several experiments to verify our theoretical claims in practical settings and address concerns regarding Assumption 4:
    1. Adversarial training of a 3-layer neural networks with varying perturbaton sizes ($\epsilon = 8/255, 12/255$ and $24/255$) and logistic loss.
    2. Adversarial training of a CNN with varying perturbation sizes ($\epsilon = 16/255$ and $32/255$) and cross-entropy loss.

    These experiments (Appendix C.1 of the revision) showed that adversarial training accuracies can achieve 100\% in all cases, which supports Assumption 4, and the adversarial normalized margins keep increasing after the separation of adversarial training examples, which further verifies claims of Theorem 5 for other loss functions and architectures.
- We further elaborated the proof of Theorem 2 to make it more clear and exact.
- We added discussion on applying Theorem 5 to deep linear network with $\ell_{\infty}$ perturbations. This indicates that previous results on deep linear network can also be derived from Theorem 5.

We have highlighted these points in the revision.

---

### Decision · Program_Chairs · 2022-01-20

**Decision:**

Accept (Poster)

**Comment:**

The paper is a nice addition to the developing theory of implicit bias in neural training. While the results are somewhat expected, the technical aspects are fairly involved due to the adversarial component.